# Phytochrome-dependent responsiveness to root-derived cytokinins enables coordinated elongation responses to combined light and nitrate cues

Pierre Gautrat [1,2] ✉, Sara Buti [1], Andrés Romanowski [1,2], Michiel Lammers[2], Sanne E. A. Matton [2], Guido Buijs[1] & Ronald Pierik [1,2] ✉

Plants growing at high densities can detect competitors through changes in the composition of light reflected by neighbours. In response to this far-red-enriched light, plants elicit adaptive shade avoidance responses for light capture, but these need to be balanced against other input signals, such as nutrient availability. Here, we investigated how Arabidopsis integrates shade and nitrate signalling. We unveiled that nitrate modulates shade avoidance via a previously unknown shade response pathway that involves root-derived *trans*-zeatin (tZ) signal and the BEE1 transcription factor as an integrator of light and cytokinin signalling. Under nitrate-sufficient conditions, tZ promotes hypocotyl elongation specifically in the presence of supplemental far-red light. This occurs via PIF transcription factors-dependent inhibition of type-A *ARRs* cytokinin response inhibitors. Our data thus reveal how plants co-regulate responses to shade cues with root-derived information about nutrient availability, and how they restrict responses to this information to specific light conditions in the shoot.

Plants are constantly challenged by various environmental fluctuations to which they can respond using their strong developmental plasticity. Aerial shoots and underground roots must cope with and adapt to constraints from very different surroundings to fulfil their function: mainly reproduction and photosynthesis for shoots, and water and nutrient uptake for roots. Environmental cues must be deciphered locally while also taking in account remote signals from other organs to ensure adequate developmental responses and tight coordination at the level of the whole organism. Such integration of local and systemic signals is mandatory to plant survival, notably in the context of foraging responses for nutrients or light[1]. Systemic communication is made possible by endogenous mobile signal molecules, such as hormones, that travel through vascular connections, and lead to different effects depending on the local context they are perceived in[2].

While responses to individual environmental stimuli and associated systemic signals are relatively well described, integration of multiple environmental signals remains poorly understood. This is typically the case for the integration of two major limiting factors in plant growth that can occur at high planting densities: nitrate availability fluctuations and shade by neighbouring plants.

Nitrate (N) is the major form of nitrogen taken up by plants from the soil, which is the most limiting nutrient for growth in agricultural soils[3]. This leads to the massive use of fertilisers, associated pollution and high economic costs[4,5]. N is both a nutrient and a signal, perceived by the NITRATE TRANSPORTERS (NRTs), that triggers a wide range of root and shoot developmental responses[6]. In brief, low N (LN) availability, favours root growth over shoot growth, whereas high N (HN) availability triggers opposite responses[7,8]. Notably, the promotion of

[1]Plant-Environment Signaling, Institute of Environmental Biology, Utrecht University, Utrecht, The Netherlands. [2]Laboratory of Molecular Biology, Wageningen University, Wageningen, The Netherlands. ✉e-mail: pierre.gautrat@gmail.com; ronald.pierik@wur.nl

shoot growth in response to HN is mediated by a specific species of cytokinin (CK), *trans*-zeatin (tZ)[9]. N triggers an increased production of tZ species in roots through the transcriptional upregulation of CK precursor biosynthesis genes, ADENOSINE PHOSPHATE-ISOPENTENYLTRANSFERASES (*IPT*s)[10], and of a gene encoding an enzyme responsible for precursor conversion into tZ species: CYTO-CHROME P450 MONOOXYGENASE 735A2 (*CYP735A2*)[11]. tZ species are then loaded into the xylem by the ATP-BINDING CASSETTE subfamily G14 (ABCG14) transporter and translocated from roots to shoots, leading to increased CK signalling and promotion of leaf number and size[12–14].

Shade by neighbouring competitors is mainly perceived by the aerial parts of the plant through a change in light quality. Indeed, as Red (R) light is mostly absorbed by leaves while Far Red (FR) light is mostly transmitted or reflected, a decreased R:FR ratio indicates the presence of overshadowing neighbours. The perception of low R:FR ratio triggers a series of developmental responses that help con-solidate light capture in dense vegetation, collectively known as the Shade Avoidance Syndrome (SAS). These responses include hypocotyl, petiole and stem elongation, upward leaf movement and early flowering[15]. The PHYTOCHROME B (phyB) R light photoreceptor is mainly responsible for this perception through its inactivation by a low R:FR ratio[16]. phyB inactivation leads to the de-repression of PHYTO-CROME INTERACTING FACTORS (PIFs) transcription factors. PIF4, PIF5 and PIF7 are particularly important for shade avoidance[17,18] and activate most of the SAS genes expression[19,20]. Low R:FR ratio per-ception in the shoots also reduces lateral root development thanks to the shoot-to-root mobile transcription factor ELONGATED HYPOCO-TYL5 (HY5)[21].

Little is known how low R:FR and N signalling integrate. On the root side, shoot-derived HY5 directly binds and upregulates *NRT2.1* expression[22] and the shade-related HY5 regulation of lateral root development is dependent on N availability[23]. On the shoot side, plants grown under LN conditions have a lower hypocotyl elongation in response to low R:FR compared to plants grown under sufficient N conditions, independently of HY5[23]. Outside of a shade context, con-stant levels of N do not affect hypocotyl length, while an upshift from LN to HN promotes elongation[24].

Here we investigated the molecular mechanisms underlying the impact of N availability on SAS, using *Arabidopsis thaliana* hypocotyl elongation as a model. Using genetic and pharmaceutical approaches, we unravelled that HN promotes hypocotyl elongation through root-derived tZ signal and canonical CK signalling. A transcriptomic approach revealed that N availability and tZ widely impact low R:FR responses. Our data suggest that the positive shade avoidance actor BR-ENHANCED EXPRESSION 1 (BEE1) is an integrator of shade and CK signalling. The hypocotyl growth-promoting effect of tZ occurred only under low R:FR conditions and not under high R:FR. The suppression of a tZ response under high R:FR conditions was shown to be lifted under low R:FR mainly depending on phyB and PIF7, through down-regulation of multiple type-A *ARABIDOPSIS RESPONSE REGULATORS* (*ARRs*), which are negative regulators of CK signalling. Together, these findings demonstrate that tZ is a novel actor promoting shade-induced elongation and that its effect on elongation is tightly gated by phyB-PIF signalling. This regulation may prevent unnecessary elongation under non-shaded conditions and allows the integration of N availability in the fine tuning of shade-induced shoot elongation.

## Results and discussion
### High Nitrate promotes shade avoidance responses through root-derived tZ signal and canonical CK signalling
We first recapitulated the differential elongation of *Arabidopsis thali-ana* hypocotyls in response to low R:FR depending on N availability[23]. Four days old Col-0 seedlings grown either under high nitrate (HN; 10 mM KNO$_3$) or low nitrate (LN; 0.2 mM KNO$_3$) regimes were

subjected for four more days to White Light (WL, high R:FR = 2.5) or White Light with supplemental Far-Red light (WL + FR, low R:FR = 0.25). While under WL conditions, N availability did not affect hypocotyl elongation, WL + FR-induced elongation was drastically lower in LN than HN regime (Fig. 1A, B). We then tested whether N availability affects shade avoidance via the established N-activated tZ systemic pathway[9]. The *cyp735a1,a2* double mutant (*cypDM*), deficient in tZ species production[11] and *abcg14*, deficient in root-to-shoot tZ species transport[12,13] both displayed a reduced elongation response to WL + FR specifically in HN conditions compared to Col-0 (Fig. 1C–E). This highlights that CYP735As and ABCG14 play a positive role in hypocotyl elongation responses to supplemental FR under HN conditions, but not under LN. The *ipt3,5,7* triple mutant, partially impaired in CK pre-cursor biosynthesis[25], seemed to have mildly reduced hypocotyl length in WL + FR, although this was not significantly different from Col-0 (Supplementary Fig. 1A). We also investigated if NIN-LIKE PROTEIN 7 (NLP7), a transcription factor partially acting upstream of the *CYP735A2* and *IPT3* upregulation by HN[26], was involved in the N-dependent control of WL + FR-induced elongation. *nlp7* indeed exhibited a reduced WL + FR-induced hypocotyl elongation, but both under HN and LN regimes, highlighting a broader role than tZ-deficient mutants that act specifically under HN conditions (Supplementary Fig. 1B). To verify whether tZ-deficient mutants are not generally impaired in hypocotyl elongation responses, etiolation experiments in the dark were performed and revealed that these mutants are capable of strong hypocotyl elongation and reached similar lengths as Col-0 (Supplementary Fig. 1C). Although CKs are typically considered to inhibit hypocotyl elongation in dark-grown seedlings, it was shown that the synthetic CK 6-Benzylaminopurine (BAP) at relatively high concentrations ($10^{-6}$ M range) indeed inhibits elongation in the dark but promotes hypocotyl growth in the light[27]. We confirmed this to be true for both BAP and tZ applications, highlighting that both CK spe-cies at $10^{-6}$ M promote hypocotyl elongation in the light, while inhi-biting it in the dark (Supplementary Fig. 1D–E). How CKs can have opposite effects on hypocotyl length in dark and light is currently unknown.

We then investigated if CK overaccumulation impacts shade avoidance using a CYTOKININ OXIDASES (CKXs) quadruple mutant, *ckx3456*, which is widely impaired in CK degradation[28]. *ckx3456* exhibited an increased elongation in response to WL + FR under both HN and LN regimes (Fig. 1F), indicating that CK overaccumulation promotes shade avoidance independently of N regime. Another approach was used to confirm this observation with the application of exogenous tZ directly on the shoots to avoid the pleiotropic pheno-type observed when applied to the whole seedling[29]. Different tZ concentrations were tested ranging from $10^{-8}$ to $10^{-7}$ M. All con-centrations exacerbated the WL + FR-induced hypocotyl elongation, while only the highest tested concentration, $10^{-7}$ M, elicited a weak increase under WL conditions (Supplementary Fig. 1F). We pursued the next experiments with the $10^{-8}$ M tZ treatment as this was the lowest concentration inducing elongation under WL + FR, reminiscent of the observations on the *ckx3456* mutant. tZ application on plants grown under LN regime also induced a stronger elongation in response to WL + FR but not under WL illumination. tZ induced similar elongation under HN and LN regimes, rescuing WL + FR hypocotyl length of LN-grown plants treated with tZ to the same length as HN-grown mock treated plants (Fig. 1G, H and Supplementary Fig. 1G). This highlights that the reduced elongation response to WL + FR in LN-grown plants is most likely due to signalling components, not nutrient limitation, with tZ being sufficient to rescue this response. tZ application to *ipt3,5,7*, *cypDM*, or *abcg14* mutants resulted in an elongation resembling tZ-treated Col-0 plants, showing that lack of tZ in the shoots of these mutants is likely causal of their phenotype (Supplementary Fig. 1H–J).

Grafting experiments were then performed to assess if CYP735As and ABCG14 are required in either shoots or roots for the WL + FR

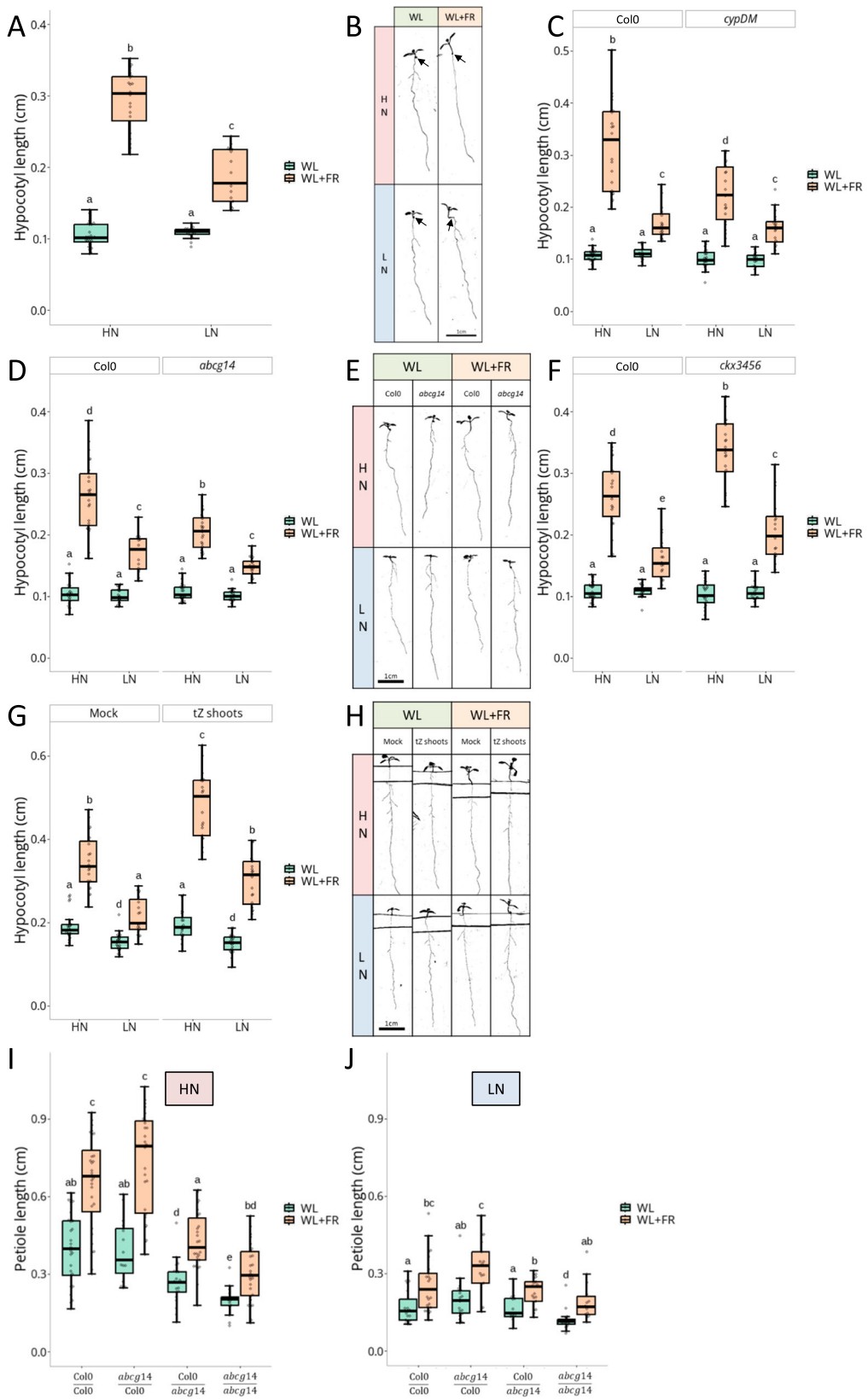

elongation response. As this required the cutting of hypocotyls for the grafts, we could not use hypocotyl length as a readout. Alternatively, we measured petiole elongation; another well-established and FR-sensitive shade avoidance trait[30,31]. *cypDM* grafts under HN and WL + FR conditions revealed that both shoot and root CYP735As are important for elongation responses, as only the *cypDM* homografts showed

significant differences to Col-0 homografts (Supplementary Fig. 1K), in accordance with previous reports that CYP735As are important in both shoots and roots for CK precursor conversion to tZ species[11]. Under HN and WL + FR conditions, *abcg14* mutation in the shoots did not lead to significant differences compared to Col-0 homografts while mutation in the roots led to a significant reduction of petiole length in WL and

**Fig. 1 | HN promotes FR light-induced hypocotyl elongation through root-derived tZ. A** Hypocotyl length in cm and representative images **B** black arrows indicate the junction between root and hypocotyl of Col-0 seedlings grown on High Nitrate (HN, 10 mM) or Low Nitrate (LN, 0.2 mM) for 4 days under White Light (WL) and then transferred 4 more days to WL or White Light + Far-Red light (WL + FR, *n* > 16 plants per condition). **C** Hypocotyl length in cm of Col-0 and *cypDM* (*n* > 15); *abcg14* **D** *n* > 15, representative images in **E** or *ckx3456* (**F** *n* > 17) grown in the same condition as **A**. **G** Hypocotyl length in cm and representative images **H** of Col-0 seedlings grown on HN or LN for 4 days under WL and then transferred 4 more days

to compartment plates treated with Mock or tZ ($10^{-8}$ M) on the shoot compartment and under WL or WL + FR (*n* > 18). **I** Petiole length in cm of hypocotyl-grafted Col-0 and *abcg14* plants after a - 10 days recovery period and treated 4 days with WL or WL + FR, under HN regime (*n* > 17), or under LN regime (**J** *n* > 14). Different letters depict statistical differences according to a Kruskal-Wallis test ($p < 0.05$). Box plots whiskers represent the intervale between the minimal value and the first quartile, and between the 4th quartile and maximal value. The box encompasses the 2nd and third quartiles, with the median indicated in the centre.

WL + FR demonstrating a role of root-localised ABCG14 in petiole elongation and shade avoidance (Fig. 1I–J). Under LN and WL + FR conditions, Col-0 homografts petiole length is similar to the other graft combinations, highlighting once again that ABCG14 is mostly important under HN conditions (Fig. 1I–J).

The involvement of canonical CK signalling actors in the regulation of hypocotyl elongation was then investigated. Double mutants for the three ARABIDOPSIS HISTIDINE KINASES (AHKs) CK receptors, *ahk2,3, ahk2,4,* and *ahk3,4*[32], and gain of function versions of these receptors, *repressor of cytokinin deficiency, rock2* and *rock3*[33], revealed a positive role of mainly AHK2 and AHK3 in shade avoidance (Supplementary Fig. 2A–C). *rock2* and *rock3* led to an increased elongation specifically under WL + FR, confirming that the positive effect of CK signalling on hypocotyl elongation requires WL + FR conditions. CK perception by AHKs leads to the activation of type-B ARRs transcription factors[34], with ARR1, ARR10, and ARR12 explaining most of the CK responses[35]. *arr1,10* showed reduced hypocotyl elongation in response to WL + FR, whereas *arr10,12* was unaffected and the triple mutant *arr1,10,12* response was completely abolished. However, this latter result needs to be interpreted cautiously since *arr1,10,12* development is drastically disturbed compared to WT plants (Supplementary Fig. 2D–G). Altogether these results highlight that root-derived tZ signal in response to HN promotes hypocotyl elongation strictly under WL + FR conditions, through the canonical AHK/type-B ARR CK signalling module.

To verify the broader significance of these findings on Arabidopsis hypocotyls, we also studied adult Arabidopsis and found that spraying tZ on Arabidopsis rosettes promoted FR-induced petiole elongation but had no effect under WL conditions (Supplementary Fig. 3A, C), similarly to what is observed on hypocotyls. However, the hyponastic response (upward leaf movement) observed in response to WL + FR was not altered by tZ application (Supplementary Fig. 3B, C), suggesting that tZ specifically affects organ elongation responses to supplemental FR. The promotion of shoot elongation is not limited to Arabidopsis as we confirmed that two other species, *Brassica rapa* (turnip) and *Solanum lycopersicum* (tomato) also display a promotion of hypocotyl length by tZ application specifically under WL + FR conditions (Supplementary Fig. 3D–G).

## N availability and tZ deficiency widely affects WL + FR transcriptomic responses

To unveil how N availability impacts shade avoidance through tZ, we conducted a transcriptomic survey of shoots from Col-0, *cypDM*, and *abcg14* plants grown under HN or LN regimes and subjected or not to WL + FR for 90 min (Fig. 2A). A Principal Component Analysis (PCA) highlighted that PC1 was explained by light conditions and PC2 by N availability (Supplementary Fig. 4A). Differentially Expressed Genes (DEGs) between WL and WL + FR conditions (WL + FR DEGs) in Col-0 were drastically affected by N availability, with approximately 52% of the upregulated genes and 61% of the downregulated genes dependent on the N regime (Fig. 2B, Supplementary Fig. 4B). Because tZ-deficient mutants play a role in shade avoidance responses under HN conditions (Fig. 1C–E), we then focused our gene expression analysis on the Col-0 WL + FR DEGs that are strictly regulated under HN regime. We compared these 369 genes with the *abcg14* and *cypDM*

WL + FR HN DEGs and found that more than half of the WL + FR HN DEGs in Col-0 were no longer regulated in these mutants (Fig. 2C and Supplementary Fig. 4C). We then further investigated the 190 genes upregulated by WL + FR requiring both HN regime and functional tZ species accumulation, by conducting a Gene Ontology (GO) enrichment analysis. No enrichment for shade avoidance related GO terms were found anymore, whereas such enrichment was present if considering all genes upregulated by WL + FR under HN conditions (Supplementary Fig. 4D), but other GO terms such as response to salicylic acid, abscisic acid, or oxidative stress were enriched (Fig. 2D). We collected mutants for five genes selected from the pool that displayed FR-induction in HN but were no longer FR-induced in LN and in the *abcg14* and *cypDM* mutants (Supplementary Fig. 4E). Although most of these mutants displayed a wildtype response to WL + FR, two knock-out alleles of *ABA- AND OSMOTIC-STRESS-INDUCIBLE RECEPTOR-LIKE CYTOSOLIC KINASE1* (*ARCK1*, a gene encoding a receptor-like cytosolic kinase negatively regulating ABA signalling[36]) indeed showed an altered shade avoidance response in HN (Supplementary Fig. 4F). This indicates how the approach taken here can identify novel genetic components of shade avoidance regulation linked to N and tZ.

We then investigated whether a direct link existed between the expression of established shade avoidance actors and CK signalling. We noticed a strong enrichment for shade avoidance GOs in the 458 WL + FR DEGs regulated independently of N availability, but not in other conditions (Fig. 2B, D, E, and Supplementary Fig. 4D). While these 458 genes were identified in a DEG list of both HN and LN conditions in response to WL + FR, we identified in a subsequent clustering analysis a small cluster of 39 genes showing a higher z-score in WL + FR Col-0 HN compared to LN or the cytokinin mutants (Fig. 2F, Supplementary Data 1A). The FR-induced upregulation of these 39 genes thus is higher in HN than in LN, and in Col-0 versus the tZ-deficient mutants. To identify the genes whose promoters are direct targets of type-B ARRs, and therefore of CK signalling, we compared this list with published type-B ARRs ChIP-Seq studies[37,38]. To avoid too many unspecific targets, we filtered these datasets by retaining only the 444 genes whose promoters were found in both studies, and were targeted by all three ARR1, ARR10 and ARR12 in Xie et al.[38] (Supplementary Data 2). Out of the 39 genes, five of their promotors were identified as direct targets of type-B ARRs (Supplementary Data 1B). We further filtered those five genes to retain the ones that showed a significant attenuation of WL + FR expression induction by N availability and tZ-deficient mutants. This led to a final list of three genes: *BEE1*, PHYTOCHROME RAPIDLY REGULATED 1 (*PAR1*), and an uncharacterised transcription factor *AT1G75490* (Fig. 2G, Supplementary Fig. 4G). Out of these three genes, only *BEE1* expression is also regulated by exogenous CK applications (Supplementary Data 1C)[38]. Interestingly, the BEE1 bHLH transcription factor and the transcriptional co-factor PAR1 interact with each other and are part of the same regulatory module[39], with BEE1 playing a positive role in SAS and PAR1 a negative role. Also, three genes from GO categories associated to CK were found in the 39 genes list, namely *CKX5*, *LOG2* and *ERF9*, but for none of them, the WL + FR-induced expression was affected by nitrate availability (Supplementary Data 1D).

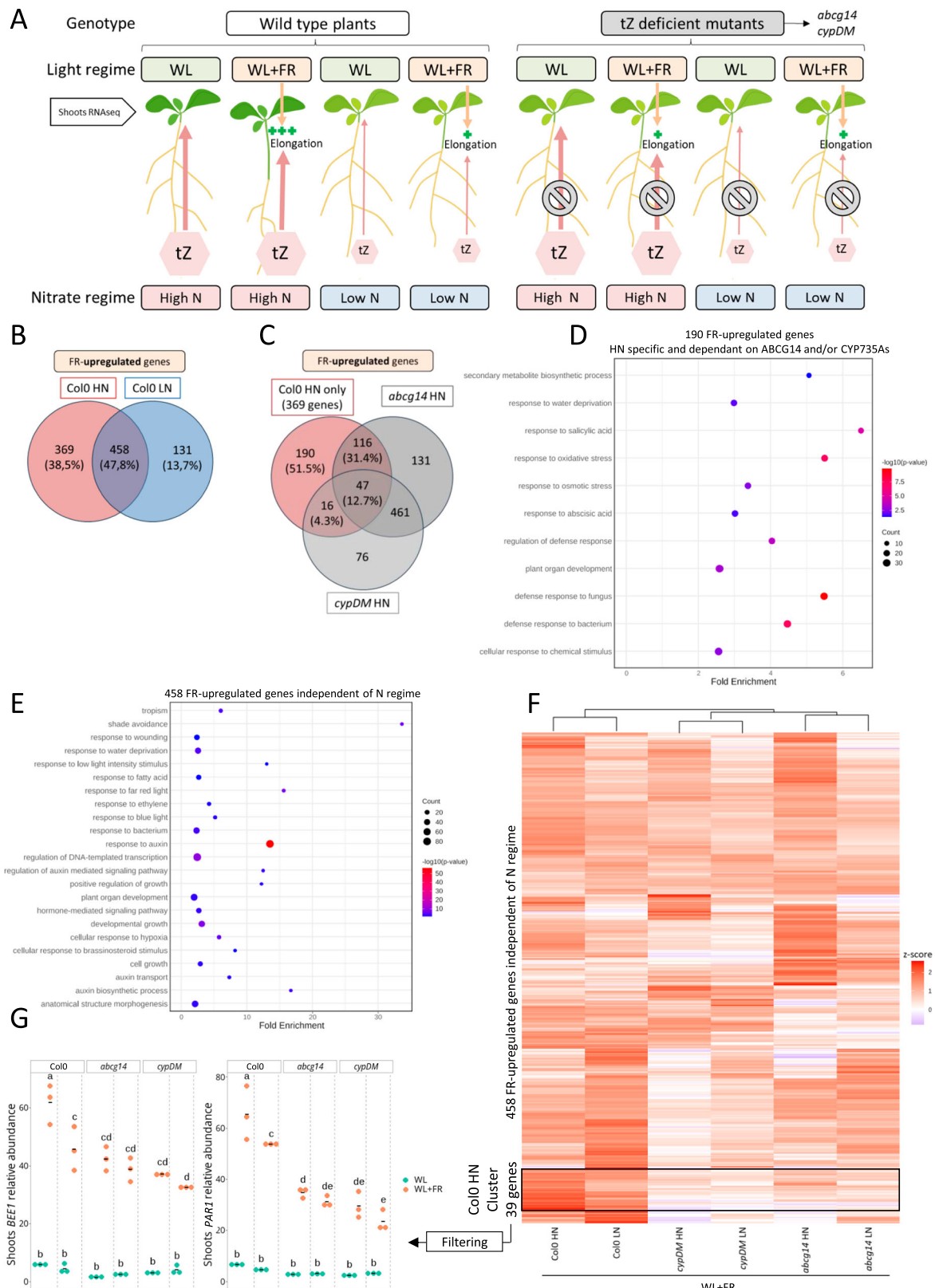

## BEE1 is an integrator of shade avoidance and CK signalling

We then focused on BEE1, a transcription factor whose promoter is a direct target of both PIF4, PIF5[19] and of the type-B ARRs ARR1, ARR10 and ARR12[37,38]. Its full expression induction by WL + FR is also dependent on N availability and functional tZ actors (Fig. 2G). To validate with another approach the link between CK signalling and BEE1 expression, we performed transactivation assays in *Nicotiana benthamiana*. This

revealed that the promoter of BEE1 (pBEE1) is transactivated by the type-B transcription factor ARR10 (Fig. 3A), confirming the positive role of CK signalling on BEE1 expression. qPCR assays on BEE1 in the type-B ARR (positive CK signalling actors) triple mutant *arr1,10,12* and type-A ARR (negative CK signalling actors) quadruple mutant *arr3,4,5,6* further strengthened this conclusion. BEE1 expression is lower in *arr1,10,12* compared to Col-0 under both WL and WL + FR conditions

**Fig. 2 | Nitrate availability and tZ deficiency widely affect FR light-induced transcriptome changes. A** Schematic of the transcriptome design. RNAseq was performed on shoots of Col-0 WT plants and tZ deficient mutants (*abcg14* and *cypDM*) grown on High Nitrate (HN, 10 mM) or Low Nitrate (LN, 0.2 mM) for 4 days under White Light (WL) and then subjected to WL or White Light + Far-Red light (WL + FR) for 90 min. Width of the red arrows depict amount of tZ root-to-shoot translocation based on already published data and "+" signs the magnitude of elongation in WL + FR. **B** Venn diagram representing the overlap of Col-0 HN vs. Col-0 LN genes upregulated by WL + FR compared to their respective WL controls (log₂ FC ≥ 1, FDR < 0.05). **C** Venn diagram representing the overlap of the 369 genes only upregulated by WL + FR in Col-0 under HN conditions and genes upregulated in *cypDM* and *abcg14* by WL + FR compared to their respective WL controls (log₂ FC ≥ 1, FDR < 0.05). **D** Bubble plot representing Gene Ontology (GO) enrichment analysis for the 190 genes whose upregulation by WL + FR depends on HN and tZ deficient mutants. GO Biological Process (BP) most specific category subclasses with a significant enrichment (Fisher's Exact test two-sided, Bonferroni corrected, *p* < 0.05) are plotted. X axis represents the fold enrichment compared to a random sample of genes. -log₁₀(*p*-value) is indicated by colours and number of genes per category is indicated by dot size. **E** Bubble plot representing GO enrichment analysis for the 458 genes commonly upregulated by WL + FR both under HN and LN conditions in Col-0. For GO BP, the most specific category subclasses with a significant enrichment (Fisher's Exact test two-sided, Bonferroni corrected, *p* < 0.05) are plotted. X axis represents the fold enrichment compared to a random sample of genes. -log₁₀(*p*-value) is indicated by colours and number of genes per category is indicated by dot size. **F** Heatmap representing the z-score (indicated by colour) of the 458 genes commonly upregulated by WL + FR both under HN and LN conditions in Col-0 across all WL + FR treated samples. Samples and genes are clustered by similarities of regulations. A gene cluster with a stronger z-score in the "Col-0 HN FR" sample than in any other sample is highlighted with a black box. **G** Normalised Count Per Million (CPM) values of *BEE1* and *PAR1* across all transcriptome samples. Each dot represents a biological replicate (pool of *n* > 20 plants) and black bars the mean of the biological replicates. Different letters depict significant differences according to a two-way ANOVA followed by a Tukey's post hoc test (*p* < 0.05).

(Supplementary Fig. 5A), and higher under WL in *arr3,4,5,6* (Supplementary Fig. 5B). *pBEE1* was also transactivated by PIF4 and PIF5, but not PIF7, confirming BEE1's already established connection to shade avoidance signalling (Supplementary Fig. 5C). However, qPCR assays in *pif* mutants revealed a dominant role of PIF7 in WL + FR-induced *BEE1* expression, pointing towards the importance of PIF7 in this regulation in Arabidopsis seedling shoots (Supplementary Fig. 5D).

A potential role of BEE1 in shade avoidance as a downstream actor of N availability and tZ was tested by isolating two *bee1* null alleles from the NASC collection (*bee1-1* N868188, and *bee1-2* N865244), that both showed a strong reduction of hypocotyl elongation in response to WL + FR compared to Col-0 under HN regime (Supplementary Fig. 6A). *bee1-1* showed no difference in hypocotyl length in WL compared to Col-0 plants, but was significantly shorter in the WL + FR treatment under HN conditions (Fig. 3B). However, the suppression of FR-induced elongation under LN conditions was more severe in Col-0 than in *bee1-1*, resulting in similar hypocotyl lengths under LN conditions for both genotypes (Fig. 3B and Supplementary Fig. 6B). This highlights that BEE1 is a positive actor in shade-induced elongation under HN conditions. *bee1-1* also showed reduced responsiveness to tZ application (Fig. 3C and Supplementary Fig. 6C). As BEE2 and BEE3 were described to act redundantly with BEE1[39,40], we assessed their expression in our transcriptome data, revealing that these two genes are upregulated by WL + FR but this does not seem to depend on N availability or functional tZ actors (Supplementary Fig. 6D–E). The triple mutant *bee123* showed a very similar response to *bee1-1* (Supplementary Fig. 6F–G). This indicates that BEE1 alone plays a strong role in shade-induced hypocotyl elongation in the context of nitrate nutrition and tZ signalling, and that the involvement of BEE2 and BEE3 is minor. We also generated a *35S:BEE1* overexpression line that exhibited a constitutively elongated hypocotyl phenotype already under WL conditions and that has reduced sensitivity to N availability (Fig. 3D, E and Supplementary Fig. 6H).

Taken together, these results show that *BEE1* is an integrator of shade avoidance and N availability cues. Its tight regulation of expression by supplemental FR light and CK signalling finely tunes hypocotyl elongation, at least partially explaining the role of tZ in shade avoidance.

### A gating effect by PIFs restricts tZ positive action on hypocotyl elongation to WL + FR through regulation of type-A *ARRs*

We then investigated why tZ and CK signalling only control hypocotyl elongation under WL + FR but not WL conditions. Indeed, changes in hypocotyl elongation in response to tZ or CK signalling alteration were only observed under WL + FR conditions, and not WL, except through high concentrations of exogenous tZ applications (Fig. 1, Supplementary Figs. 1, 2). We therefore suspected that the change of response

between WL and WL + FR conditions is due to a modification of CK signalling itself. Using the Two Component signalling Sensor new (TCSn) TCSn:GUS reporter line for CK signalling[41], we confirmed that there is an increased CK response in the hypocotyl of plants subjected to WL + FR compared to WL (Supplementary Fig. 7A, B).

As auxin plays a major role in promoting hypocotyl elongation and its production, signalling, and transport are strongly increased in response to WL + FR[42], we tested if elevated auxin levels would explain why tZ only promotes elongation in supplemental FR light conditions. Co-applications of tZ and IAA (Indole-3 Acetic Acid, 10⁻⁶ M) revealed that IAA applications were not sufficient to allow a positive effect of tZ on hypocotyl elongation under WL conditions, while co-applications lead to additive effects under WL + FR conditions (Supplementary Fig. 8A, B). The transcriptomic data generated in this study revealed that the most enriched GO term from commonly downregulated genes by WL + FR under both N regimes in Col-0 was "cytokinin-activated signalling pathway" (Supplementary Figs. 4B, 9A). The genes associated with this category were mainly type-A *ARRs*, namely *ARR4*, *ARR5*, *ARR6*, *ARR7* and *ARR15*. Type-A ARRs suppress CK signalling by competing/interacting with type-B ARRs, but their transcription is also upregulated by type-B ARRs as part of a negative feedback loop[43]. A heatmap of all type-A *ARRs* detected in the transcriptome highlighted that all of them have a lower z-score in Col-0 plants grown under WL + FR conditions compared to WL (Supplementary Fig. 9B), indicating downregulation by WL + FR. Out of the 5 type-A *ARRs* that are significantly downregulated by WL + FR, we focused on the ones with the most interesting expression patterns: *ARR4*; expressed mostly in the vasculature, and *ARR5* and *ARR6*; mostly expressed in the hypocotyl[44]. Functional type-B ARRs are necessary for basic expression levels of these three *ARRs*, and their regulation by WL + FR (Supplementary Fig. 9C). Phenotyping of the *arr4*, *arr5* and *arr6* mutants showed a negative role in FR-induced hypocotyl elongation for ARR5 and ARR6 (Fig. 4A), revealing their involvement in shade avoidance.

We then asked how type-A *ARRs* are regulated by supplemental FR light, and if this occurred in a phytochrome-dependent manner. qPCR on Col-0 and *phyB* photoreceptor mutant revealed that the strong transcriptional downregulation of type-A *ARRs* by WL + FR is dependent on phyB, and that WL levels of type-A *ARRs* expression are low in the constitutively shade avoiding *phyB* mutant for *ARR6* and *ARR4*, and intermediate for *ARR5* (Fig. 4B and Supplementary Fig. 10A). tZ application on the *phyB* mutant led to hypocotyl elongation already under WL conditions, in contrast with the absence of effect in Col-0 (Fig. 4C and D). We then investigated if phyB-mediated regulation of type-A *ARRs* expression occurs through the canonical shade avoidance PIFs: PIF4, PIF5 and PIF7. qPCR on Col-0, *pif45*, *pif7* and *pif457* mutants showed that indeed *ARR4*, *ARR5*, and *ARR6* downregulation by WL + FR is dependent on PIFs, with PIF7 playing the

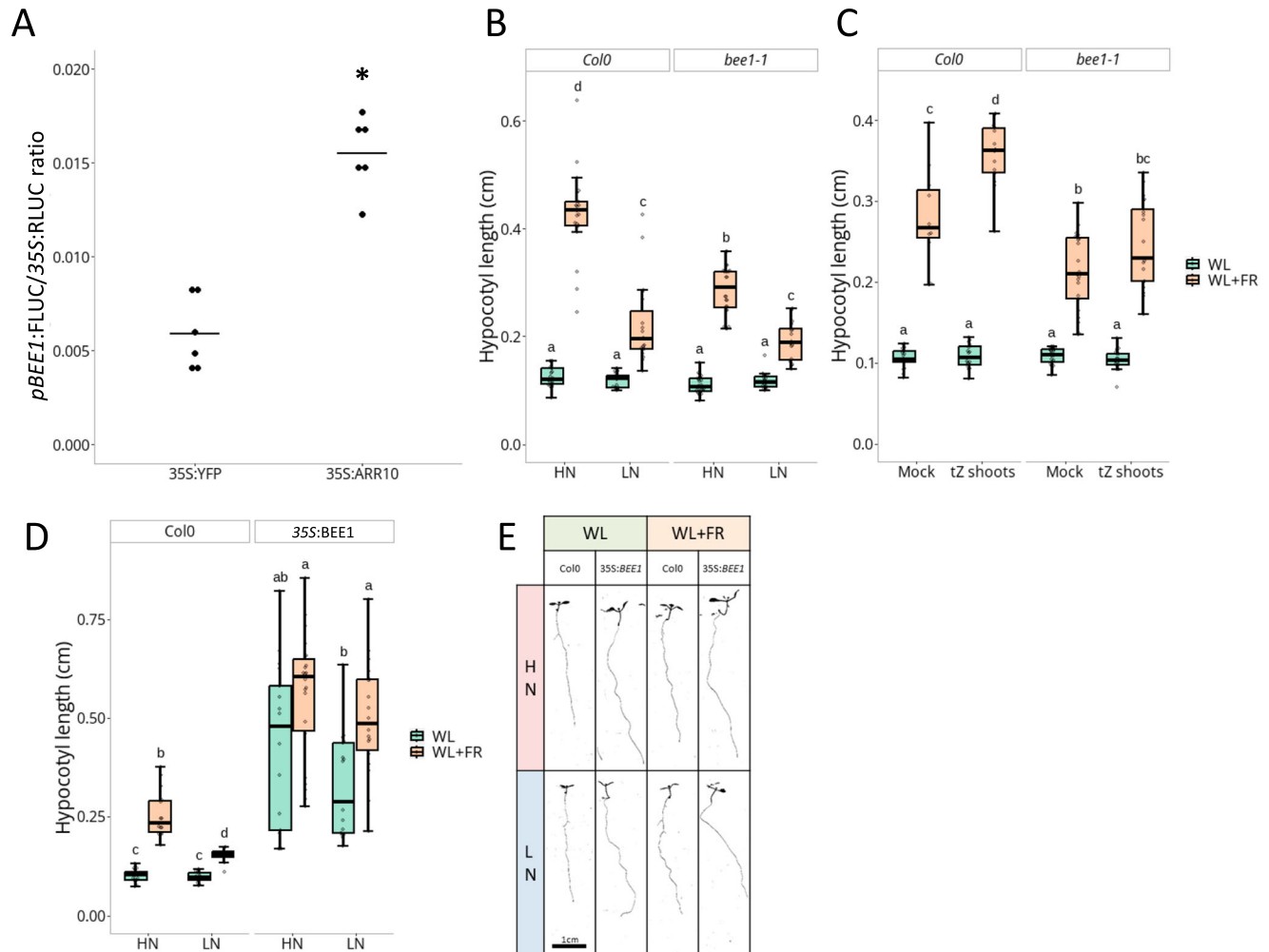

**Fig. 3 | Cytokinin signalling activates *BEE1* expression to regulate nitrate and tZ-dependent shade avoidance responses. A** Transactivation assay in *Nicotiana benthamiana*. A construct expressing *pBEE1:FireflyLUC* and *p35S:RenillaLUC* was co-infiltrated with a construct expressing *p35S:YFP* (baseline control) or *p35S:ARR10*. The FireflyLUC reporter activity was expressed ratiometrically to the RenillaLUC internal control. Each dot represents a biological replicate (*n* = 6) and black bars the mean of the biological replicates. The asterisk depicts a significant difference according to a two-way ANOVA followed by a Tukey's post hoc test (*p* < 0.05). **B** Hypocotyl length in cm of Col-0 and *bee1-1* seedlings grown on High Nitrate (HN, 10 mM) or Low Nitrate (LN, 0.2 mM) for 4 days under White Light (WL) and then transferred 4 more days to WL or White Light + Far-Red light (WL + FR, *n* > 16 plants per condition). **C** Hypocotyl length in cm of Col-0 and *bee1-1* seedlings grown for 4 days under WL and then transferred 4 more days to compartment plates treated with Mock or tZ ($10^{-8}$ M) on the shoot compartment and under WL or WL + FR (*n* > 11). **D** Hypocotyl length in cm and representative images **E** of Col-0 and *35S:BEE1* seedlings (n > 8) grown in the same condition as **B**. For **B**–**D**, different letters depict statistical differences according to a Kruskal-Wallis test (*p* < 0.05). Box plots whiskers represent the intervale between the minimal value and the first quartile, and between the 4th quartile and maximal value. The box encompasses the 2nd and third quartiles, with the median indicated in the centre.

strongest role (Fig. 4E, Supplementary Fig. 10B). Concomitantly, while *pif45* still shows increased elongation in response to WL + FR and tZ compared to WL + FR alone, *pif7* has severely reduced response to tZ application, and *pif457* is even entirely irresponsive to tZ applications (Fig. 4F), indicating that tZ action on hypocotyl elongation requires PIF4, PIF5 and especially PIF7.

Finally, we investigated if PIF7 can interfere with type-B ARR-mediated activation of type-A *ARR* expression using transactivation assays. As expected, overexpression of the type-B ARR ARR10 leads to transactivation of the type-A *ARRs* promoters *pARR4*, *pARR5* and p*ARR6* (Fig. 4G, Supplementary Fig. 10C and D). Remarkably, PIF7 overexpression strongly suppressed the ARR10-mediated activation of *pARR4* and *pARR5* and even inactivated basic *pARR4* and *pARR5* (Fig. 4G, Supplementary Fig. 10C). We could not confirm this for p*ARR6*, but as the proximal *ARR6* promoter is very short (662 bp), we might have missed the DNA motif responsible for the trans-inactivation by PIF7, that might be distal (Supplementary Fig. 10D). Our data indicate that WL + FR inhibits type-A *ARRs* expression via PIFs,

thus likely sensitising the plants to CK by reducing most of the CK signalling negative feedback loop. We substantiated this hypothesis by showing that *arr4* and *arr3,4,5,6* mutants already gain a tZ-induced hypocotyl elongation in WL, while wildtype Col-0 cannot respond to tZ (Supplementary Fig. 10E).

Altogether, this work revealed that HN availability positively regulates WL + FR-triggered hypocotyl elongation through the transport of root-derived tZ signal, involving ABCG14, and canonical CK signalling actors. N availability and tZ deficiency largely impact WL + FR transcriptome changes, including the upregulation of the positive SAS actor *BEE1*, directly depending on ARR10, that partially explains how tZ promotes shade avoidance. Promotion of hypocotyl elongation by tZ was found to be controlled by phyB activity, involving PIFs that transcriptionally suppress some of the type-A *ARRs*, which are inhibitors of CK signalling and shade avoidance (Fig. 5).

Early studies identified a direct interaction between phyB and ARR4 proteins, leading to a reciprocal stabilisation and regulation of R light signalling[45,46]. The present work identified an additional transcriptional

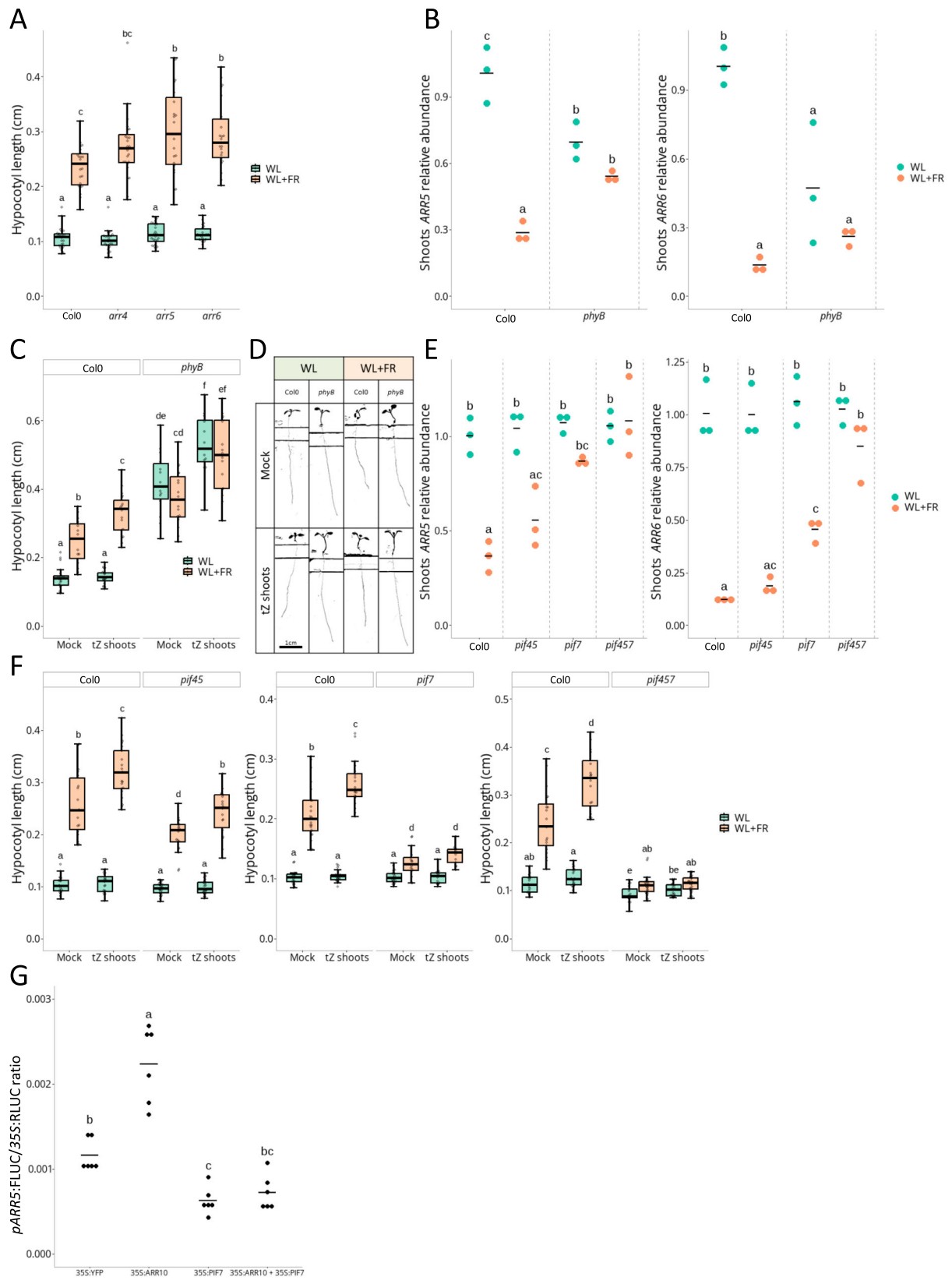

regulation of type-A *ARRs* depending on WL + FR and PIFs, showing that a multilayered regulation of type-A *ARRs* by light signalling exists and contributes to a tight regulation of hypocotyl growth.

It is important to keep in mind that we used stable N treatments throughout the experiments presented here. A recent study on changes in N availability (called downshifts and upshifts) revealed that an N upshift can promote hypocotyl elongation under standard WL condition, unlike the observations that stable HN and LN yield similar hypocotyl lengths. N upshift acts via NLP7, PIF4 and a SMALL AUXIN UP-RNA (SAUR)-dependent auxin response to promote hypocotyl length[24]. This is a different pathway from the auxin-independent mechanism identified here, and the existence of both likely allows to

**Fig. 4 | WL + FR elicits transcriptional downregulation of type-A *ARRs*, in a phyB and PIF7-dependent manner, allowing tZ action on hypocotyl elongation.** **A** Hypocotyl length in cm of Col-0, *arr4*, *arr5* and *arr6* grown for 4 days under White Light (WL) and then transferred 4 more days to WL or White Light + Far-Red light (WL + FR, *n* > 21 plants per condition). **B** Shoots *ARR5* and *ARR6* transcripts relative abundance measured by qPCR in Col-0 and *phyB* seedlings grown for 4 days under WL and then transferred for 90 min to WL or WL + FR light. Each dot represents a biological replicate (pool of *n* > 20 plants) and black bars the mean of the biological replicates. **C** Hypocotyl length in cm and representative images **D** of Col-0 and *phyB* seedlings grown for 4 days under WL and then transferred 4 more days to compartment plates treated with mock or tZ ($10^{-8}$ M) on the shoot compartment and under WL or WL + FR (*n* > 12). **E** Shoots *ARR5* and *ARR6* transcripts relative abundance measured by qPCR in Col-0 *pif45*, *pif7* and *pif457* seedlings grown as described in **B**. Each dot represents a biological replicate (pool of *n* > 20 plants) and black bars the mean of the biological replicates. **F** Hypocotyl length in cm of Col-0

and *pif45* (n > 17); *pif7* (n > 13) or *pif457* (*n* > 16) seedlings grown as described in **C**. **G** Transactivation assay in *Nicotiana benthamiana*. A construct expressing *pARR5:FireflyLUC* and *p35S:RenillaLUC* was co-infiltrated with a construct expressing *p35S:YFP* (baseline control), *p35S:ARR10* or *p35S:PIF7* or both *p35S:ARR10* and *p35S:PIF7*. The FireflyLUC reporter activity was expressed ratiometrically to the RenillaLUC internal control. Each dot represents a biological replicate (*n* = 6) and black bars the mean of the biological replicates. Different letters depict significant differences according to a two-way ANOVA followed by a Tukey's post hoc test (*p* < 0.05). For phenotyping experiments (**A**, **C** and **F**) different letters depict statistical differences according to a Kruskal-Wallis test (*p* < 0.05). For qPCR and transactivation experiments (**B**, **E** and **G**), different letters depict significant differences according to a two-way ANOVA followed by a Tukey's post hoc test (*p* < 0.05). Box plots whiskers represent the intervale between the minimal value and the first quartile, and between the 4th quartile and maximal value. The box encompasses the 2nd and third quartiles, with the median indicated in the centre.

respond specifically to dynamic nitrogen availabilities and different light cues. It would be interesting to investigate if such signalling pathways are also involved in shade avoidance responses in roots. While it was previously shown that shade avoidance-mediated repression of lateral root growth is inhibited by LN[23], we here demonstrate that shade avoidance-induced hypocotyl elongation is enhanced under sufficient N supply. Together this indicates that the level of shoot/root ratio adjustment induced by shade is tightly coordinated with N provision.

Shade avoidance responses in Arabidopsis are typically accompanied by reduced leaf blade expansion[47,48]. Since we observe here that CK promotes hypocotyl elongation in response to WL + FR, while CK is also well known to promote leaf expansion in response to nitrate[9,49], this raises the question how in low R:FR CK can promote elongation yet leaf expansion can be suppressed. An earlier study showed that the cytokinin oxidase *CKX5*, catabolizing bioactive CK, can be upregulated in response to WL + FR, thus reducing CK levels and suppressing leaf expansion[50]. As this regulation occurs mostly in the leaf and cotyledons, as opposed to the elongating tissues such as hypocotyls and petioles[50–52], this could create spatially explicit variations in bioactive CK levels, while supplemental FR would still sensitise the entire shoots to their local CK levels via the downregulation of type-A *ARR* genes as shown here.

A previous study revealed that both tZ, and its precursor *trans*-zeatin riboside (tZR) are transported from root to shoot, and impact different shoot development processes[53]. It therefore remains to be clarified what form of tZ species transport is important for shade avoidance responses, and if specific LONELY GUY (LOG) enzymes, involved in the conversion of tZ to tZR, play a role in hypocotyl elongation.

Although we focused our study on the root-derived cytokinin tZ species, there are also CK that are synthesised locally in the shoot, mostly iP types[9], that potentially may modulate shade responses. Now that we have shown a clear importance of CK signalling for shade avoidance, it will be interesting to investigate broader roles of CK in shade responses in plants. Such studies could also include alternative CK response routes, such as via CYTOKININ RESPONSE FACTORS (CRFs) that were recently described to regulate a transcriptional increase of *PIN* auxin transporters in the context of leaf expansion[49]. Since PINs are crucial regulators of shade avoidance[30,51,54,55], this could represent another way for CK to interact with shade avoidance.

We show here that tZ is an important positive regulator of shade avoidance under sufficient N, in different developmental contexts and species, while also being a well-established promotor of plant growth and yield in response to sufficient N[9]. Shade avoidance responses, although adaptive in natural vegetations of mixed species, are considered to negatively impact crop yield, in part due to reduced investments in harvestable tissues[15]. Untangling yield promotion and shade avoidance stimulation responses linked to CK may help engineer

crops with optimized shade avoidance traits, while keeping a high yield potential.

## Methods

### Plant material, growth conditions and phenotyping method

All *Arabidopsis thaliana* genotypes used in this study are in the Col-0 background: *cyp735a1,cyp735a2* (*cypDM*)[11], *abcg14*[13], *ckx3456*[28], *ipt3,5,7*[25], *nlp7-1*[56], *ahk2,3*[32], *ahk2,4*[32], *ahk3,4*[32], *rock2*[33], *rock3*[33], *arr1-3,arr10-5*[35], *arr10-5,arr12-1*[35], *arr1-3,arr10-5,arr12-1*[35], *arck1-1*[36] SALK insertion[57] NASC code N678530, *arck1-2*[36] SALK insertion NASC code N662142, *bglu45-1* SALK insertion N25150, *bglu45-2* SALK insertion N662198, *ext3-1*[58], *ext3-2*[58], *upb1-1*[59], *sen1-1*[60] SALK insertion N665464, *sen1-2* GABI-Kat insertion[61] N888009, *bee123*[40], *35S:BEE1* (generated in this study, see "Cloning and transformation procedures"), *bee1-1* SAIL insertion[62] N868188, *bee1-2* WiscDsLox insertion[63] N865244, *TCSn:GUS*[41], *arr4*[44], *arr5*[44], *arr6*[44], *arr3,4,5,6*[44], *phyB-9*[16] rid of the *ven4* mutation, *pif4-101,pif5-1*[64], *pif7-1*[65] and *pif4-101,pif5-1,pif7-1*[47]. *Brassica rapa* (turnip) "Groene" and *Solanum lycopersicum* (tomato) "Money-maker" varieties were used.

Seeds were surface sterilised with chlorine gas or with ethanol and bleach (tomato) and sowed in vitro on a modified 1/2MS medium where the nitrogen source was replaced with either 10 mM $KNO_3$ (HN) or 0.2 mM $KNO_3$ + 9.8 mM KCL (LN). The other nutrients composition and concentration were as follow: $CaCL_2$ 1.5 mM, $MgSO_4$ 0.75 mM, $KH_2PO_4$ 0.625 mM, Fe(III)Na-EDTA 50 µM, $MnSO_4$ 50 µM, $H_3BO_3$ 50 µM, $ZnSO_4$ 15 µM, $Na_2Mo_4$ 0.5 µM and $CuSO_4$ 0.1 µM. Medium was supplemented with 1 g/L 2-(N-morpholino)ethanesulfonic acid (MES), 1% Plant Agar (Duchefa Biochemie) and pH adjusted to 5.8 with KOH (no sucrose or vitamins used). For most in-vitro experiments, after a 3 days stratification period (4 °C, in the dark, in the dark but 20 °C for tomato), seeds were transferred to a growth chamber with controlled conditions: long days photoperiod (16 h light, 8 h dark), PAR = 150, White Light (WL, R:FR = 2.5), 70% humidity. In vitro plates were placed in a modified D-root system[66], where roots are shielded from the light and only shoots are exposed. After 4 days under WL conditions (two days for turnip and tomato), homogeneous seedlings were selected and transferred to new plates: either plain plates containing HN or LN for most experiments, or plates divided in two parts with a trench to allow pharmaceutical treatment of the shoot compartment with either tZ (Duchefa Biochemie), at the different concentrations mentioned, or IAA (Duchefa Biochemie) at $10^{-6}$ M, and their respective mock treatments. Seedlings were then subjected to either WL conditions or WL + FR (R:FR = 0.25) for 4 days.

For etiolation experiments, seeds received a 2 h WL pulse after stratification and were transferred at 20 °C in the dark for 4 more days before phenotyping. For comparison of dark vs. light response of seedlings to CK, an etiolation experiment or a four day growth assay in WL of seedlings growing on HN medium supplemented with tZ $10^{-6}$ M or BAP $10^{-6}$ M (ThermoFischer Scientific) was performed.

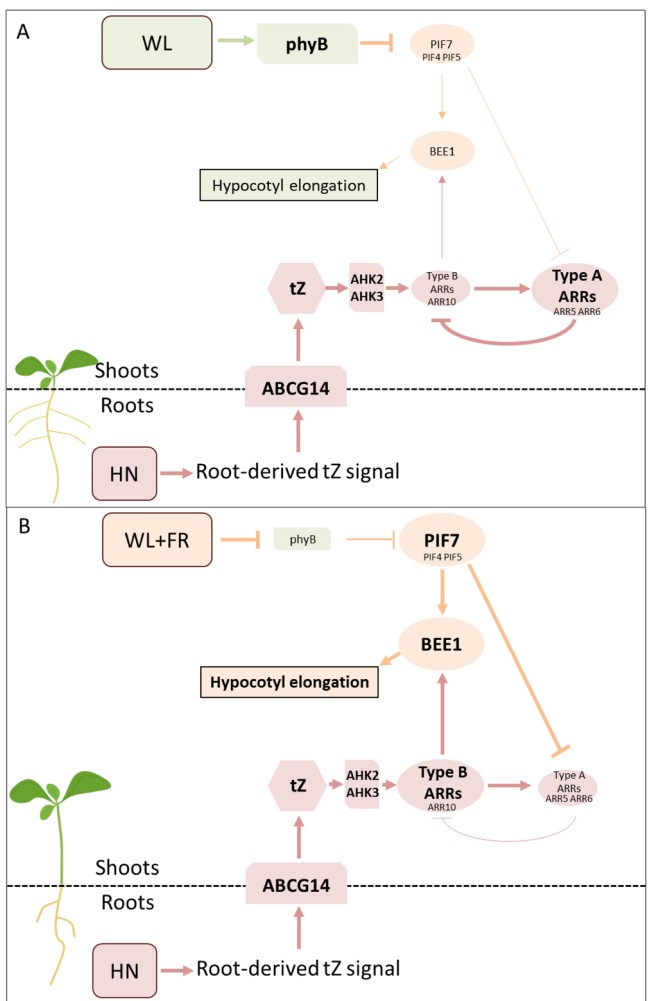

**Fig. 5 | Integration of light and nitrate signalling regulates hypocotyl elongation responses to neighbour proximity cues.** Model of Arabidopsis hypocotyl elongation response under HN regime and contrasting light qualities: either high R:FR (WL, **A**) or low R:FR ratio (WL + FR, **B**). Under WL, phyB is active, repressing PIF transcription factors activity, preventing hypocotyl elongation. Under WL + FR conditions, phyB is inactivated, leading to the de-repression of PIFs and shade avoidance responses, including hypocotyl elongation. In response to HN, root-derived tZ signal is translocated to the shoots through the ABCG14 transporter. Specifically under WL + FR conditions, tZ is then promoting hypocotyl elongation through the canonical AHK/type-B signalling module, and upregulation of the *BEE1* positive shade avoidance actor. This specificity of action is allowed by PIFs, particularly PIF7, via downregulation of the type-A *ARRs* negative CK actors, sensitising hypocotyls to tZ and allowing their elongation. This leads to the integration of light and N signalling to fine tune hypocotyl elongation. Plain arrows represent a positive regulation and broken lines a negative regulation. The size of the boxes and lines thickness depict their prevalence. Actors in orange are predominantly described as shade avoidance actors, in green active in WL and in red in response to sufficient nitrate nutrition. Only regulations directly relevant to the manuscript are presented, and other regulations are voluntary omitted for clarity.

At the end of in-vitro experiments, plates were scanned with an Epson V800 scanner at the end of the experiment. Hypocotyl or petiole length was measured using the SmartRoot[67] ImageJ plugin.

For soil grown experiments, 12-d-old Col-0 grown in WL were photographed at ZT4, transferred to WL or WL + FR and treated with either a mock or $5.10^{-8}$ M tZ spray. 24 h later, plants were photographed again. Petiole length and angles of the two first true leaves were measured using ImageJ at both times T0 and T24, and the delta values calculated for each individual plants ($\Delta$=T24−T0).

## Grafting procedure

Hypocotyl grafts were performed on 5 to 6 days old seedlings. Scions and root stocks were cut with a razorblade towards the middle of the hypocotyl, assembled in a 2 mm long, 0.3 mm diameter silicone tube (HelixMark, Freudenberg Medical) and placed in vitro on a Hybond N membrane (Amersham Pharmacia biotech) over HN medium. Grafted plants were grown for 6 days before selection of successful grafts, removal of the silicon tubes, excision of adventitious roots if present, and transfer to new HN or LN medium plates. After 4 more days of recovery, grafted plants were transferred to WL or WL + FR for 4 days. Petioles were measured instead of hypocotyls as they were cut, and plants too old to elicit proper hypocotyl elongation responses.

## Tissue sampling, RNA extractions, cDNA synthesis and qPCR experiments

Around 20 shoots of 4 days old seedlings grown under WL and HN conditions were harvested and flash frozen in liquid nitrogen immediately after a 90 min WL or WL + FR treatment at ZT8. Tissues were ground to powder using a bead basher and RNAs>200nt were extracted using a Qiagen RNAeasy Minikit. Residual DNA was removed using the on column Qiagen RNAse-free DNAse, following the manufacturer's instructions. cDNA synthesis was performed on 5 ng/μL RNA using the SuperScript III Reverse Transcriptase (ThermoFisher Scientific), following the manufacturer's instructions. Two cDNA syntheses per sample were generated as technical replicates. qPCRs were performed in a 5 μL reaction mix using 2 μL cDNA, qPCRBIO SybrGreen Blue Mix (PCRBIOSYSTEMS) and efficient primers specifically amplifying target genes (listed in Supplementary Table 1). 40 amplification cycles were performed using a CFX Opus 384 Real-Time PCR System (BIO-RAD), followed by a melting curve. *APT1* and *PP2AA3* were used as reference genes (Supplementary Table 1). Relative expression of samples was done using the ΔΔCt method[68], with samples from Col-0 grown under WL conditions as the set reference.

## High-throughput mRNA sequencing

Total RNA was extracted from around 20 shoots of 4 days old Col-0, *abcg14* and *cypDM* plants treated with either WL or WL + FR light for 90 min, and grown either with HN or LN content, as described above. Samples were then sent to Novogene for quality check and sequencing. Briefly, RNA quality was checked with the Agilent 2100 Bioanalyzer G2939A system. Between 20 and 80 ng of RNA was used to prepare 36 libraries using the Novogene NGS Stranded RNA Library Prep Set (PT044); following the manufacturer's protocol. Sample libraries were pooled equimolar and sequenced on a Novaseq 6000 (Illumina) with 150 bp pair-end reads, yielding between 40 and 63 million raw reads per sample.

## Processing of RNA sequencing reads

Raw sequence quality was checked with FastQC 0.11.8 (http://www.bioinformatics.babraham.ac.uk/projects/fastqc/) and reads were trimmed and quality checked using TrimGalore 0.6.6 (https://github.com/FelixKrueger/TrimGalore), which is a wrapper for Cutadapt 2.1[69] and FastQC 0.11.8, with default parameters and a set 'AGATCGGAAGAGC', to eliminate adaptor contamination from the PE reads. Trimmed reads were aligned against the *A. thaliana* genome (TAIR10) with HISAT2 2.2.1[70] with default parameters, except in the case of the maximum intron length parameter, which was set at 5,000. Count tables for the different feature levels were obtained from bam files using custom R scripts in R 4.2.3 and considering the AtRTD3 transcriptome[71]. Briefly, for this purpose, we used the 'ASpli::gbCounts()' function of ASpli package version 2.6.0[72], which uses the GenomicFeatures Bioconductor package[73].

## Differential gene expression analysis

Counts per million (CPMs) were then obtained with the cpm() function of edgeR v3.38.4[74,75]. Genes with a minimum of 10 cpm in all samples within one group were considered expressed and included in the analysis. This resulted in 24,823 out of 42,924 genes (60.65 %) being considered for further downstream analysis. Before determining Fold Changes and significance, the counts were normalised (TMM, trimmed mean of M-values) by correcting for differences in library sizes and compositional biases. Fold Changes were subsequently determined with the Bioconductor R package edgeR v3.38.4. Differentially Expressed Genes (DEGs) were then estimated based on the response to treatment and/or nitrogen content for each genotype. The resulting *p-values* were adjusted for multiple comparisons with the Benjamini-Hochberg method yielding a False Discovery Rate (FDR) criterion. Genes with FDR values lower than 0.05 were considered differentially expressed. Detailed information about the statistics for each graph can be found in the respective figure legends.

## Gene Ontology enrichment and hierarchically clustered heatmaps analyses

Gene Ontology (GO) enrichment analysis was performed using the Panther classification system[76], version 18.0. 27,436 *Arabidopsis thaliana* genes and GO categories from the Panther v18.0 (DOI: 10.5281/zenodo.7942786) were used. Most specific GO subclasses showing significant enrichment according to a Fisher's Exact test followed by a Bonferroni correction ($\alpha < 0.05$) were selected for graphs. Bubble plots were generated using RStudio. The genes identified in the transcriptome as both ARR-B targets and regulated differentially by shade depending on CK signalling and N availability were filtered using the following GO CK categories to test if they were known CK actors: GO:0009736 cytokinin-activated signalling pathway, GO:0009690 cytokinin metabolic process, GO:0000160 phosphorelay signal transduction system, GO:0080036 regulation of cytokinin-activated signalling pathway, GO:0009735 response to cytokinin.

Hierarchically clustered heatmaps were generated using the ggdendroplot R package (https://github.com/NicolasH2/ggdendroplot).

## Cloning and transformation procedures

Cloning was performed using the Gateway method[77] with BP and LR clonase (Thermo Fisher Scientific) to generate the constructs used in this study. Primers used for cloning are listed in Supplementary Table 1. The donor vector used to clone the sequence of interest was pDONR207. The expression vectors finally obtained were: for transactivation assays, pEarleyGate 100 ARR10 CDS (*35S:*ARR10, full length CDS), pEarleyGate 103 PIF4 (*35S:*PIF4-GFP), pEarleyGate 103 PIF5, pEarleyGate 103 PIF7, pGreenII 0800-LUC *pBEE1* (*pBEE1:*FLUC + *35S:*RLUC), pGreenII 0800-LUC *pARR4*, pGreenII 0800-LUC *pARR5*, pGreenII 0800-LUC *pARR6*; and for stable transformants generation, pEarleyGate 100 BEE1 CDS. The pB7WGY2 (*35S:*YFP) construct used as a baseline control for transactivation assays was already available[78]. Cloning was carried out using thermocompetent Top10 *Escherichia coli*. Electrocompetent AGL-1 *Agrobacterium tumefaciens* cells were transformed with the above-mentioned expression vectors and used for agroinfiltrations and stable transformant generations. Stable transformants were obtained by flower dipping as described previously[79].

## Transactivation assays in Nicotiana benthamiana

*Nicotiana benthamiana* leaves of the same developmental stage were co-agroinfiltrated with the Agrobacterium carrying the expression vector with the promoter of interest driving FireflyLUC expression and *35S:*RenillaLUC, and the Agrobacterium carrying the expression vector(s) overexpressing transcription factors. Infiltration was made using MMA liquid medium (2.2 g/l MS, 1.95 g/L MES, 20 g/L sucrose, 200 μM

acetosyringone, pH=5.6) with an $OD_{600} = 0.2$ for each Agrobacterium. After 2 days, leaf discs at the site of infiltration were harvested and flash frozen. Samples were grinded to powder and processed using the Dual-Glo Luciferase Assay System (Promega) kit, following manufacturer's instructions. Dual luminescence measurements were performed using a GloMax 96 Microplate Luminometer (Promega) apparatus. Two technical replicates were conducted per sample. The results are presented as the ratio of FLUC/RLUC fluorescence, with RLUC as the internal standard for transformation efficiency.

## GUS staining protocol

Four days old *TCSn:GUS* seedlings grown under WL and on HN medium were subjected to a 90 min WL or WL + FR treatment at ZT8. Whole seedlings were instantly fixed in 90% acetone at −20 °C for 20 min. Samples were then washed two times using a GUS staining solution without X-Gluc (0.1 M PhospH-Pi pH=7, 10 mM EDTA, 1 mM $K_3Fe(CN)_6$, 1 mM $K_4Fe(CN)_6*3H_2O$). After washing, samples were incubated for 90 min at 37 °C in a GUS staining solution supplied with 0.5 mg/mL X-Gluc (Biosynth). The staining reaction was stopped with 3:1 acetic acid/ethanol for 60 min. Samples were cleared overnight in 70% ethanol before imaging using a M205FA stereomicroscope (Leica).

## Statistical analysis

Phenotyping experiments data were analysed using a Kruskal-Wallis statistical test ($\alpha < 0.05$), DEGs were selected with a log2FC >=1 and an FDR < 0.05, and all other experiments tested with a two-way ANOVA followed by a Tukey's HSD test ($\alpha < 0.05$), expect if indicated otherwise. All statistical analyses were performed with RStudio.

## Reporting summary

Further information on research design is available in the Nature Portfolio Reporting Summary linked to this article.

## Data availability

Raw sequences (.fastq files) and processed.bam files used in this paper have been deposited in the ArrayExpress[80] database at EMBL-EBI (www.ebi.ac.uk/arrayexpress) under accession number E-MTAB-13638. Pre-processed data is also readily available for download in Zenodo (https://doi.org/10.5281/zenodo.10371355). All custom R scripts are available at https://github.com/aromanowski/shade_N_CK and can be downloaded from Zenodo (doi pending). Source data are provided with this paper.

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

## Acknowledgements

We thank Sandrine Ruffel for providing the *abcg14*, *cypDM*, *ipt3,5,7* and *nlp7-1* seeds; Thomas Schmülling for providing the *ahk2,3*, *ahk2,4*, *ahk3,4*, *rock2* and *rock3* seeds; and Gwyneth Ingram for providing the *ext3-1* and *ext3-2* seeds. We thank Lotte van der Krabben, Muthanna Biddanda, Lucila Salvatore and Gabriele Panicucci for technical assistance. We thank Alysha Somer and Leonardo Jo for helping with scripts writing. We thank Kirsten ten Tusscher and Leonardo Jo for critical reading of the manuscript, and three anonymous reviewers for very helpful feedback on a draft of this manuscript. This work was funded by EMBO Postdoctoral Fellowship Programme (ALTF 828-2020) to P.G., European Union's Horizon 2020 research and innovation programme under the Marie Sklodowska-Curie grant agreement No 101026742 to A.R., and the Netherlands Organisation for Scientific Research ENW Vici grant 865.17.002 to R.P.

## Author contributions

P.G. designed the research with additional input from R.P.; P.G., S.B., M.L., S.M., and G.B. performed the research; A.R. and P.G. analysed the data; P.G. and R.P. wrote the manuscript.

## Competing interests

The authors declare no competing interests.
