## [Peer Review File · Nature Communications]

REVIEWER COMMENTS

Reviewer #1 (Remarks to the Author):

Light and nutrient availability are critical environmental factors modulating all aspects of plant development growth. However, how plants integrate light and nutrient signals remains poorly understood. Here, the authors investigated how shade and nitrate signaling integrate to modulate plant growth. The authors provide solid genetic evidence demonstrating that high nitrate promotes hypocotyl elongation via root-derived transZeatin via canonical cytokinin signaling. Using transcriptomic analysis, they showed that nitrate availability and transZeatin profoundly affect the shade avoidance responses at the transcriptomic level. These experiments identified BEE1 as an integrator of shade and cytokinin signaling. Supporting this conclusion, the triple mutant *bee123* showed a shorter hypocotyl phenotype only under the shade and high-nitrate condition. Moreover, the authors provide genetic evidence that the trans-Zeatin response is repressed under normal light conditions, explaining the specificity of the transZeatin and cytokinin signaling under combined shade and high nitrate conditions.

Overall, this is a novel and beautiful study that elucidates novel genetic elements in the genetic circuitry whereby shade and nitrate signaling integrate to modulate plant growth. The manuscript is very well written. I only have a couple of minor suggestions.

1. From the transcriptomic analysis, BEE1 was identified as a potential integrator of shade and cytokinin signaling. However, the triple *bee123* mutant was used for the subsequent phenotypic analysis. Please discuss whether the levels of BEE2 and BEE3 were regulated by shade and cytokinin signaling and what the phenotype of the *bee1* single mutant was.

2. The authors previously showed that auxin transport (from shoots) is critical for the shade avoidance response. The results here indicated that co-application of IAA was insufficient to allow a positive effect of transZeatin on hypocotyl elongation under white-light conditions (Sup. Figure 6A,B). Could the authors elaborate on whether IAA could still modulate the transZeatin effects under shade conditions?

Reviewer #2 (Remarks to the Author):

Activation of de novo cytokinin biosynthesis and the root-to-shoot transport via xylem in response to nitrogen nutrition and phytochrome-PIF-mediated regulation of hypocotyl elongation in response to red/far-red light ratio (shade avoidance response) have been intensively studied independently. However, little attention has been paid to the interplay between the two mechanisms.

In this manuscript, the authors found the mechanism that controls hypocotyl elongation by integrating light-quality sensing and nitrate-nutrient sensing and also clarified the importance of root-derived cytokinin signal as one of the integration signaling molecules. They also identified a key gene, BEE1 (a positive actor in FR-induced hypocotyl elongation under HN condition), which acts downstream of the cytokinin signaling pathway.

This paper is an original work that proposes an integrated model for two major limiting factors in plant growth: variation in nitrate availability and changes in light quality in densely planted plants.

I would like to address my concerns as follows.

Major points:

1: In the grafting experiments (Fig 1I, Sup Fig S1I), the authors measured only petiole length at WL+FR. As the author mentioned, petiole length is a known indicator of avoidance response, but without data on WL, HN, and LN, it is difficult to accurately interpret the values measured. I understand the laboriousness of the preparation of many grafted plants, but the full set of experimental data will corroborate the authors' claims more firmly.

2: Was the ARR10-expressing construct used in Fig 3A the full length or did it exclude the negatively acting receiver domain? If it is full-length, was cytokinin applied during the assay? Was the signaling driven by endogenous cytokinins?

When full-length B-type ARR is introduced into a cell, its function is dependent on cytokinin signaling in the host cell. So, the effect often could not be detected or very weak.

Constitutively active form eliminating the amino-terminal side receiver domain is often used for this purpose. What was the case in this study?

3: The model diagram in Fig. 5 and in the text uses the phrase "root borne tZ," but the authors should be more careful with the expression. According to Osugi et al. (2007), both active (tZ) and precursor (tZR) forms of cytokinins are transported through the xylem, and they act on different traits in shoot growth (Nat Plants 3: 17112). The tZR acts after a process of conversion to the active form by LOG. It is unclear which molecular species is primarily responsible for the regulation of hypocotyl elongation revealed in this study.

To clarify this point, the authors should ideally examine the effect of mutation of LOG gene family expressed in the hypocotyl. However, in this paper, identifying the actual cytokinin molecular species acting in hypocotyl elongation might not be a prerequisite for publication.

However, it should be clearly stated in the discussion as a remaining issue even if that experiment is not performed. At least in the model diagram in Figure 5 and in the text, it should be a "root-derived tZ signal" to avoid confusion.

4: The authors' experiments demonstrated that pBEE1 was trans-activated by PIF4 and PIF5, but not by PIF7 (Sup. Fig 4A). On the other hand, PIF7 played the strongest role in the downregulation of ARR4, ARR5, and ARR6 by WL+FR (see Figure 4E, Sup. Fig 8B). This role differentiation among the PIFs is not well discussed, nor is depicted in the model diagram in Fig 5. In Fig 5, PIF7 appears to have a dominant role in BEE1 expression. This ambiguity in drawing the model diagram is the most confusing point to the reader.

The authors should be more explicit about the different roles among the PIFs.

Minor points

5: It is difficult to see the plots in Fig 2G, 4B, 4E, Sup Fig 3E, etc. I suggest the authors to improve them in a better way.

6: L227 and other parts: The use of the adverb "surprisingly" and "strikingly" should be avoided as much as possible. The data presented in a scientific paper are always new findings in principle.

Reviewer #3 (Remarks to the Author):

This manuscript reports on the integration of light and nitrate signalling to regulate Arabidopsis hypocotyl elongation in the context of shade avoidance, which is a response to a high proportion of far red light in the light spectrum. Interestingly, one of these factors, nitrate availability, is sensed by the root, while the other factor R/FR ratio, is perceived by the shoot, and both signals are integrated to regulate growth in the tissue in between, the hypocotyl of the germinating seedling. Nitrate availability in the root is transformed into a hormonal signal, the cytokinin (CK) trans-zeatin. High nitrate promotes hypocotyl elongation via increased root CK synthesis, but this becomes effective only under high FR and not under white light. In this way nitrate availability regulates a typical shade avoidance growth response. The authors study the underlying mechanism and show that PHYB and

PIF7 downregulate A-type response regulators, which are negative regulators of CK signaling, so that CK can become active. Furthermore, they identify BEE1 as an integrator of this crosstalk.

I have few points to be considered.

trans-zeatin is proposed to be the relevant cytokinin acting on hypocotyl elongation. However, there are publications (Sakakibara group) showing that zeatin riboside is the main transported cytokinin from root to shoot and that trans-zeatin and trans-zeatin riboside regulate different developmental processes in the shoot (e.g. Osugi et al., 2017). The authors have analysed different CK biosynthesis mutants and should draw the attention to their conclusions on tZ/tZR.

TransZeatin should be written *trans-zeatin* (with *trans* written italic)

Others have reported before that hypocotyl elongation is inhibited by cytokinin mimicking the action of light in the dark (see e.g. Cary et al., Plant Phys 107, 1075, 1995; Cortleven et al., J Exp Bot. 2019, 70, 165-178, the same B-type ARRs are involved as here). This suggested that the effects of CK and light on hypocotyl elongation are additive (see Su and Howell, Plant Phys. 108, 1423-1430). How can the completely opposite effect of CK under the conditions used here be explained?

The ChIP data of Zubo et al., 2017 and Xie et al., 2020 are used to identify targets of B-type ARRs among the 31 regulated genes. However, the published data set are very large with thousands of genes and thus a very high probability to find unspecific targets. Numerous known CK target genes (including primary response genes) are not included in the lists. This analysis should be refined (for example by including meta-data on CK-regulated genes) and described in more detail.

The model suggests that PIF proteins promote BEE1 activity through direct transcriptional regulation and indirectly through suppression of A-type ARRs resulting in increased B-Type activity targeting BEE1. Direct evidence for both of these pathways are only provided by experiments using transient expression in tobacco protoplasts. The authors should think of additional ways to provide evidence for these mechanisms. What are BEE1 expression levels in a *pif* mutant? In *arr* mutants? Is the high ARR5 expression in the *pif7* mutant dependent on ARR10?

Given the high functional redundancy of A-type ARR genes (often even multiple mutants do not have a phenotype) it is a bit surprising that mutations in single A-type ARR genes have an effect. are there other examples that could be cited?

For transient expression in protoplast using multiple proteins at the same time, it would be important to show that both these proteins are being expressed.

In the model, the spatial organization of the different factors is a bit difficult to interpret. Cytokinin (tZ or also tZR?) comes from the root and acts in the hypocotyl, while phyB perceives the light signal in leaves (which is not formally shown). Downstream of tZ and phyB act type-B ARR_s and PIF7 on BEE1 but where this located is unclear. In addition, the model shows a BEE1-independent action of PIF7 on hypocotyl elongation, the source of this information is unclear. How is the signal transmitted from PIF7 (leaves) to the hypocotyl? PhyB might be moved a bit up to fit in the scheme. The WL situation (without FR) should be shown for comparison.

At the end the authors propose that the results of their research will help to develop better plants for agriculture. To support this statement it would be good to discuss the relevance of this a bit more in detail. The best would be to show that other shade avoidance responses are regulated in a similar manner as has been done for some grafting experiments, the elongation of the hypocotyl of *Arabidopsis* is not of overwhelming agricultural importance.

General:

We would like to thank the editor and three reviewers for their very thoughtful and constructive feedback. We believe this has strongly improved the rigour and broad reach of our manuscript. Since some of the suggestions prompted us to perform rather challenging assays (such as larger-scale grafting experiments), we needed the maximum revision time to return this revised manuscript.

Reviewer #1 (Remarks to the Author):

Light and nutrient availability are critical environmental factors modulating all aspects of plant development growth. However, how plants integrate light and nutrient signals remains poorly understood. Here, the authors investigated how shade and nitrate signaling integrate to modulate plant growth. The authors provide solid genetic evidence demonstrating that high nitrate promotes hypocotyl elongation via root-derived transZeatin via canonical cytokinin signaling. Using transcriptomic analysis, they showed that nitrate availability and transZeatin profoundly affect the shade avoidance responses at the transcriptomic level. These experiments identified BEE1 as an integrator of shade and cytokinin signaling. Supporting this conclusion, the triple mutant *bee123* showed a shorter hypocotyl phenotype only under the shade and high-nitrate condition. Moreover, the authors provide genetic evidence that the trans-Zeatin response is repressed under normal light conditions, explaining the specificity of the transZeatin and cytokinin signaling under combined shade and high nitrate conditions.

Overall, this is a novel and beautiful study that elucidates novel genetic elements in the genetic circuitry whereby shade and nitrate signaling integrate to modulate plant growth. The manuscript is very well written. I only have a couple of minor suggestions.

1. From the transcriptomic analysis, BEE1 was identified as a potential integrator of shade and cytokinin signaling. However, the triple *bee123* mutant was used for the subsequent phenotypic analysis. Please discuss whether the levels of BEE2 and BEE3 were regulated by shade and cytokinin signaling and what the phenotype of the *bee1* single mutant was.

This is a great observation and we decided to perform a new set of experiments using one of the two *bee1* alleles that we studied for their responsiveness to WL+FR under high nitrate conditions (Sup. Figure 6A, L228-231): we phenotyped the *bee1-1* mutant both under HN vs LN regimes and tested its response to tZ. The obtained results were similar to the ones observed with the *bee123* triple mutant: *bee1-1* is elongating less in response to WL+FR than Col-0 under HN regime but is similar under LN (Figure 3B, Sup. Figure 6B), and is less responsive to tZ (Figure 3C, Sup. Figure 6C).

We also provided the *BEE2* and *BEE3* expression data from the transcriptome we performed (Sup. Figure 6D-E, L237-240), which reveals that those genes are upregulated by WL+FR, but this regulation does not seem to consistently rely on nitrate availability or tZ actors.

We therefore modified our conclusion, and put more emphasis on BEE1 only, as BEE2 and BEE3 seem to play only a rather minor role in shade avoidance responses in our experimental setup (L240-242). We moved the *bee123* data to supplementary figures (Sup. Figure 6F and G), and

replaced it with the new *bee1-1* data (Figure 3B and C), and modified the text accordingly (L231-236).

2. The authors previously showed that auxin transport (from shoots) is critical for the shade avoidance response. The results here indicated that co-application of IAA was insufficient to allow a positive effect of transZeatin on hypocotyl elongation under white-light conditions (Sup. Figure 6A,B). Could the authors elaborate on whether IAA could still modulate the transZeatin effects under shade conditions?

We added a sentence (L263-264) to better describe Sup. Figure 8A and highlight that co-application of IAA and tZ under WL+FR conditions lead to additive effects on hypocotyl elongation. We, thus, do not see clear indications for modulation of the tZ treatment effects by IAA.

Reviewer #2 (Remarks to the Author):

Activation of de novo cytokinin biosynthesis and the root-to-shoot transport via xylem in response to nitrogen nutrition and phytochrome-PIF-mediated regulation of hypocotyl elongation in response to red/far-red light ratio (shade avoidance response) have been intensively studied independently. However, little attention has been paid to the interplay between the two mechanisms.

In this manuscript, the authors found the mechanism that controls hypocotyl elongation by integrating light-quality sensing and nitrate-nutrient sensing and also clarified the importance of root-derived cytokinin signal as one of the integration signaling molecules. They also identified a key gene, *BEE1* (a positive actor in FR-induced hypocotyl elongation under HN condition), which acts downstream of the cytokinin signaling pathway.

This paper is an original work that proposes an integrated model for two major limiting factors in plant growth: variation in nitrate availability and changes in light quality in densely planted plants.

I would like to address my concerns as follows.

Major points:

1: In the grafting experiments (Fig 1I, Sup Fig S1I), the authors measured only petiole length at WL+FR. As the author mentioned, petiole length is a known indicator of avoidance response, but without data on WL, HN, and LN, it is difficult to accurately interpret the values measured. I understand the laboriousness of the preparation of many grafted plants, but the full set of experimental data will corroborate the authors' claims more firmly.

This is an excellent point, but indeed doing the full set of grafting experiment as proposed is incredibly laborious. Nevertheless, we performed the full factorial set of grafting experiments as suggested by the reviewer. As the *cypDM* heterografts were not different from the Col-0 homografts controls (Sup. Figure 1K), we focused our efforts on redoing experiments involving *abcg14* grafts. The new *abcg14* grafting experiments (Figure 1 I, J) confirmed the results we previously observed under WL+FR HN conditions: no difference between Col-0 homografts and *abcg14*/Col-0 (shoot/root), but a shorter petiole length for Col-0/*abcg14* and *abcg14* homografts, pointing again towards the importance of ABCG14 in the roots for petiole length

phenotype. New information arises from those experiments as well: root *abcg14* mutation also leads to significantly lower petiole length under WL conditions, ABCG14 therefore likely plays a broader role in petiole than in hypocotyl elongation. Under LN conditions, there is no difference between the Col-0 homograft and any other graft combination under WL+FR, highlighting once again that ABCG14 is mostly important under HN conditions. We therefore removed the previous Figure 1I and J, updated it with the new data and changed the text accordingly to the new results (L139-144).

2: Was the ARR10-expressing construct used in Fig 3A the full length or did it exclude the negatively acting receiver domain? If it is full-length, was cytokinin applied during the assay? Was the signaling driven by endogenous cytokinins?

When full-length B-type ARR is introduced into a cell, its function is dependent on cytokinin signaling in the host cell. So, the effect often could not be detected or very weak.

Constitutively active form eliminating the amino-terminal side receiver domain is often used for this purpose. What was the case in this study?

The ARR10 full length CDS driven by a 35S promoter was used for the transactivation assays in tobacco, and not a constitutively active form. It has now been clarified in the material and methods (L464-465). No cytokinin was applied during the assay, so it is safe to assume that the signalling was driven by the tobacco endogenous cytokinin and that overexpressing ARR10 CDS was sufficient to further transactivate its targets in our experimental setup.

3: The model diagram in Fig. 5 and in the text uses the phrase "root borne tZ," but the authors should be more careful with the expression. According to Osugi et al. (2007), both active (tZ) and precursor (tZR) forms of cytokinins are transported through the xylem, and they act on different traits in shoot growth (Nat Plants 3: 17112). The tZR acts after a process of conversion to the active form by LOG. It is unclear which molecular species is primarily responsible for the regulation of hypocotyl elongation revealed in this study.

To clarify this point, the authors should ideally examine the effect of mutation of LOG gene family expressed in the hypocotyl. However, in this paper, identifying the actual cytokinin molecular species acting in hypocotyl elongation might not be a prerequisite for publication.

However, it should be clearly stated in the discussion as a remaining issue even if that experiment is not performed. At least in the model diagram in Figure 5 and in the text, it should be a "root-derived tZ signal" to avoid confusion.

Thank you for catching this and we apologise for the over-simplification. Every occurrence where "tZ" was used in an ambiguous way that could have been read as "tZ" or "tZR" has now been replaced by "tZ signal" or "tZ species". We also added a new part in the discussion (L336-340) to acknowledge that the specific mobile tZ species involved in shade avoidance remains to be identified, together with a potential role of LOGs in shade avoidance.

4: The authors' experiments demonstrated that pBEE1 was trans-activated by PIF4 and PIF5, but not by PIF7 (Sup. Fig 4A). On the other hand, PIF7 played the strongest role in the downregulation of ARR4, ARR5, and ARR6 by WL+FR (see Figure 4E, Sup. Fig 8B). This role differentiation among the PIFs is not well discussed, nor is depicted in the model diagram in Fig 5. In Fig 5, PIF7

appears to have a dominant role in *BEE1* expression. This ambiguity in drawing the model diagram is the most confusing point to the reader.

The authors should be more explicit about the different roles among the PIFs.

This is a very helpful remark, and we agree with respect to the ambiguity. In answering a comment from Reviewer 3 in their first point, we measured *BEE1* expression in Arabidopsis seedling shoots of *pif4,5*, *pif7* and *pif4,5,7* by qPCR (Sup. Figure 5D). In these experiments we found that out of these three PIFs, PIF7 has the strongest impact on *BEE1* expression in response to WL+FR at seedling stage. The activation of *pBEE1* by PIF4 and PIF5 as compared to PIF7 in our transactivation assays is therefore puzzling and might be related to tissue-specific context, co-expression of interacting proteins, expression in a heterologous system or still other reasons. We adjusted the text to describe the new data requested by Reviewer 3 and discuss the relevance of the transactivation assays compared to qPCR in Arabidopsis seedling shoots (L225-227). This new evidence corroborates the central role of PIF7 in *BEE1* expression modulation under WL+FR, just like it is the dominant PIF for hypocotyl elongation (Figure 4F) and *A-ARR* expression modulation (Figure 4E). We therefore kept this aspect of the model, but as indicated now better discuss the role of the different PIFs.

Minor points

5: It is difficult to see the plots in Fig 2G, 4B, 4E, Sup Fig 3E, etc. I suggest the authors to improve them in a better way.

Thank you for identifying this. We modified all expression data plots to make them more readable by using bigger dots with more contrasting colours, and separating the different mutants used with dotted lines.

6: L227 and other parts: The use of the adverb "surprisingly" and "strikingly" should be avoided as much as possible. The data presented in a scientific paper are always new findings in principle.

Although we feel that these adverbs can improve the reading experience, we also agree they are superfluous. We, therefore, removed all occurrences of "surprisingly" and "strikingly" in the text.

Reviewer #3 (Remarks to the Author):

This manuscript reports on the integration of light and nitrate signalling to regulate Arabidopsis hypocotyl elongation in the context of shade avoidance, which is a response to a high proportion of far red light in the light spectrum. Interestingly, one of these factors, nitrate availability, is sensed by the root, while the other factor R/FR ratio, is perceived by the shoot, and both signals are integrated to regulate growth in the tissue in between, the hypocotyl of the germinating seedling. Nitrate availability in the root is transformed into a hormonal signal, the cytokinin (CK) trans-zeatin. High nitrate promotes hypocotyl elongation via increased root CK synthesis, but this becomes effective only under high FR and not under white light. In this way nitrate availability regulates a typical shade avoidance growth response. The authors study the underlying mechanism and show that PHYB and PIF7 downregulate A-type response regulators,

which are negative regulators of CK signaling, so that CK can become active. Furthermore, they identify BEE1 as an integrator of this crosstalk.

I have few points to be considered.

trans-zeatin is proposed to be the relevant cytokinin acting on hypocotyl elongation. However, there are publications (Sakakibara group) showing that zeatin riboside is the main transported cytokinin from root to shoot and that trans-zeatin and trans-zeatin riboside regulate different developmental processes in the shoot (e.g. Osugi et al., 2017). The authors have analysed different CK biosynthesis mutants and should draw the attention to their conclusions on tZ/tZR.

This relevant issue was also raised by Reviewer 2 in their 3rd comment. We again apologize for the over-simplification and have fixed all occurrences where tZ was ambiguously used. We also added a discussion about this very topic (L336-340).

TransZeatin should be written trans-zeatin (with trans written italic)

Thank you for the remark, we fixed all occurrences where *trans*-zeatin was misspelled.

Others have reported before that hypocotyl elongation is inhibited by cytokinin mimicking the action of light in the dark (see e.g. Cary et al., Plant Phys 107, 1075, 1995; Cortleven et al., J Exp Bot. 2019, 70, 165-178, the same B-type ARR_s are involved as here). This suggested that the effects of CK and light on hypocotyl elongation are additive (see Su and Howell, Plant Phys. 108, 1423-1430). How can the completely opposite effect of CK under the conditions used here be explained?

Thank you for pointing us to the Cortleven et al (2019) study, where the authors discovered that while relatively high concentration of BAP suppresses hypocotyl elongation in the dark, it does the opposite in white light: hypocotyl length promotion (their Figure 8). While the mechanism explaining this switch of action is still unclear and highly interesting, this would really require a study on its own. Inspired by the reviewer comment and the mentioned paper, we performed a new set of experiments to test whether we could recapitulate the results from Cortleven et al., 2019, and if tZ is having a similar effect as BAP on hypocotyl elongation. Using HN plates supplemented with either 10⁻⁶ M BAP or 10⁻⁶ M tZ, we measured the hypocotyls of plants grown in the dark or in white light. We obtained similar results as Cortleven et al., 2019, with high CK concentrations inhibiting etiolation in the dark but promoting hypocotyl growth in WL (Sup Figure 1D and E). We have now addressed this matter in the text (L106-112).

The ChIP data of Zubo et al., 2017 and Xie et al., 2020 are used to identify targets of B-type ARR_s among the 31 regulated genes. However, the published data set are very large with thousands of genes and thus a very high probability to find unspecific targets. Numerous known CK target genes (including primary response genes) are not included in the lists. This analysis should be refined (for example by including meta-data on CK-regulated genes) and described in more

detail.

We apologize for the lack of clarity on how our analysis was done. To avoid having too many unspecific targets of B-type ARRs, we first thoroughly filtered the Zubo et al., 2017 and Xie et al., 2018 datasets by retaining only the genes that were found in both studies and that were targets of both ARR1, ARR10 and ARR12 in Xie et al., 2018. This way, we narrowed down the thousands of genes from these studies to a list of 444 genes. This is now better described in the text (L200-202).

To address this reviewer comment, we have done two new subsequent analyses to both narrow down even further the B-type target genes list, and to include all known CK target genes. To narrow down, we added another filter on the 444 genes we already isolated from the ChIP datasets, by retaining only the genes that were also shown to be regulated by exogenous CK applications, lowering the number to 53 genes. When comparing this list with the 39 genes of interest, only *CKX5* and *BEE1* were found (Sup. Table 1C). To test whether any known CK target genes missing from the ChIP datasets are still present in our list, we compared the genes present in all CK GO categories (GO numbers specified in material and methods L454-457) to this list. This led to the identification of 3 genes: *CKX5*, *LOG2* and *ERF9* (Sup. Table 1D), but none of them were differently expressed depending on nitrate availability under WL+FR (Figure EX1). Altogether, these new analyses reinforce the idea that *BEE1* is a prime candidate as an integrator of shade and CK signalling in the context of different nitrate regimes.

Figure EX1: *CKX5*, *ERF9* and *LOG2* CPM values

Normalized Count Per Million (CPM) values of *CKX5*, *ERF9* and *LOG2* across all transcriptome samples. Each dot represents a biological replicate (pool of n>20 plants) and black bars the mean of the biological replicates. Different letters depict significant differences according to a two-way ANOVA followed by a Tukey's post hoc test (p<0.05).

The model suggests that PIF proteins promote *BEE1* activity through direct transcriptional regulation and indirectly through suppression of A-type ARRs resulting in increased B-Type activity targeting *BEE1*. Direct evidence for both of these pathways are only provided by experiments using transient expression in tobacco protoplasts. The authors should think of additional ways to provide evidence for these mechanisms. What are *BEE1* expression levels in a *pif* mutant? In *arr* mutants? Is the high *ARR5* expression in the *pif7* mutant dependent on *ARR10*?

To address this comment, we first assessed by qPCR *BEE1* expression under WL vs WL+FR in the B-type ARR triple mutant *arr1,10,12* (Sup. Figure 5A), in the A-type ARR quadruple mutant *arr3,4,5,6* (Sup. Figure 5B); and in the *pif4,5*, *pif7*, and *pif4,5,7* mutants (Sup. Figure 5D). This revealed that *BEE1* expression is lower in *arr1,10,12* than in Col-0 under both WL and WL+FR conditions, while its expression is higher in *arr3,4,5,6* under WL, confirming the suggestion that was made in the model. Induction of *BEE1* expression by WL+FR was also shown to be mostly dependent on PIF7. Text has been modified to include these new results (L220-227).

The high expression of *ARR5* in *pif7* under low R:FR indicates that the strong downregulation by WL+FR in WT relies on PIF7. To study if regulation of *ARR5* by WL+FR, is dependent on B-type ARRs, we tested its expression, together with *ARR4* and *ARR6*, in the *arr1,10,12* mutant subjected to WL or WL+FR (Sup. Figure 9C). Expression of the three A-type ARRs was either very low, or not detectable in *arr1,10,12* already under WL conditions, and just as low in WL+FR conditions. This shows how B-type ARRs are necessary for basal expression levels of A-type ARRs, and therefore also their subsequent regulation by WL+FR. The text has been updated to describe these new results (L274-275).

Given the high functional redundancy of A-type ARR genes (often even multiple mutants do not have a phenotype) it is a bit surprising that mutations in single A-type ARR genes have an effect. are there other examples that could be cited?

There are a few examples in literature of a single A-type ARR mutation leading to different phenotypes from a WT in particular environmental conditions. For instance, *arr5*, *arr6* and *arr7* single mutants confer a higher survival rate to cold stress than Col-0 (Jeon et al., 2010, DOI: 10.1074/jbc.M109.096644). More relevant to the topic of the present manuscript, *arr4* single mutant was found to have an increased hypocotyl elongation response to continuous red light compared to WT plants (Mira-Rorado et al., 2007 DOI: 10.1093/jxb/erm087 and Chi et al., 2016 DOI: 10.1073/pnas.1601724113). Mira-Rorado et al., 2007 is already cited in the text (L310).

For transient expression in protoplast using multiple proteins at the same time, it would be important to show that both these proteins are being expressed.

To confirm there is expression of all proteins when infiltrated at the same time, we performed the same experiment as in Figure 4G, but now harvested the transfected *Nicotiana benthamiana* leaf disks, for RNA extractions and performed qPCRs on *ARR10* and *PIF7*, the two transcription factors we were aiming to overexpress (primers used listed in Sup. Table 2). The data are provided below (Figure EX2) and indicate that both *PIF7* and *ARR10* are well expressed when infiltrated, both in isolation and in combination, and neither of them displays reduced expression when combined with the other.

Figure EX2: Expression level of *ARR10* and *PIF7* in tobacco leaf disks

ARR10 and *PIF7* transcripts relative abundance compared to *NbActin* measured by qPCR in *Nicotiana benthamiana* leaf disks where a construct expressing *pARR5:FireflyLUC* and *p35S:RenillaLUC* was co-infiltrated with a construct expressing *p35S:YFP*, *p35S:ARR10*, *p35S:PIF7* or both *p35S:ARR10* and *p35S:PIF7*. Each dot represents a biological replicate (n=4) and black bars the mean of the biological replicates. Different letters depict significant differences according to a two-way ANOVA followed by a Tukey's post hoc test ($p < 0.05$).

In the model, the spatial organization of the different factors is a bit difficult to interpret. Cytokinin (tZ or also tZR?) comes from the root and acts in the hypocotyl, while phyB perceives the light signal in leaves (which is not formally shown). Downstream of tZ and phyB act type-B ARRs and PIF7 on BEE1 but where this located is unclear. In addition, the model shows a BEE1-independent action of PIF7 on hypocotyl elongation, the source of this information is unclear. How is the signal transmitted from PIF7 (leaves) to the hypocotyl? PhyB might be moved a bit up to fit in the scheme. The WL situation (without FR) should be shown for comparison.

We are very grateful for these comments and apologize for the lack of clarity. The scheme of a plant was meant for illustration purposes, and not to highlight regulations in different shoot organs. Our present experiments do not allow to understand where specifically in the shoots are the regulations occurring, even though this would be a very interesting future topic. We thus modified the scheme to make sure we don't give the impression that we have such spatially-resolved knowledge.

We also clarified that we do not yet know which tZ species are transported from root to shoot (also requested by Reviewer 2 in their 3rd comment), by replacing "Root-derived tZ" with "Root-derived tZ signal".

The BEE1-independent action of PIF7 on hypocotyl elongation was added to acknowledge the fact that PIF7 was previously identified as a major regulator of hypocotyl elongation, through regulation of many more factors than just *BEE1* (see for example Willige et al., 2021 DOI: 10.1038/s41588-021-00882-3). However, we understand it is not related to the story presented here and could create confusion. We therefore removed the arrow from PIF7 directly pointing to hypocotyl elongation. This is now explicit in the figure legend that only published regulations of direct relevance for our work, and our own work, is presented in the model.

We also now separated the scheme in two parts, to compare the WL vs the WL+FR situation (Figure 5) as per the reviewer's request.

At the end the authors propose that the results of their research will help to develop better plants for agriculture. To support this statement it would be good to discuss the relevance of this a bit more in detail. The best would be to show that other shade avoidance responses are regulated in a similar manner as has been done for some grafting experiments, the elongation of the hypocotyl of *Arabidopsis* is not of overwhelming agricultural importance.

Although a bit outside the scope of our work, we found this a very inspiring suggestion and decided to perform two sets of new experiments to widen the scope of this manuscript:

- As suggested by Reviewer 3, we assessed the modulation by tZ of two other shade avoidance traits of older *Arabidopsis* plants grown in soil: petiole elongation and hyponasty (upward leaf movement). This new data revealed that moderate concentration of tZ ($5 \cdot 10^{-8}M$) sprayed on 12d old *Arabidopsis* rosettes promotes petiole elongation only under WL+FR, in a similar fashion as hypocotyl length. However, tZ did not affect leaf hyponasty response to WL+FR and might therefore have a specific role on global elongation, and not differential elongation of the petiole (Sup. Figure 3A-C). The text (L158-163) and the material and methods (L391-394) have been updated to describe the new experiments and results.
- We also tested in two different crops, turnip (*Brassica rapa*) and tomato (*Solanum lycopersicum*), if tZ is impacting hypocotyl elongation responses to WL+FR. Similar to *Arabidopsis*, both species showed a specific promotion of hypocotyl length when treated with tZ, only under WL+FR conditions, and not WL (Sup. Figure 3D-G). The text (L163-166) and the material and methods (L366-379) have been updated to describe the new experiments and results.

We therefore added a short sentence in the discussion to emphasize that tZ seems to play a role in shade avoidance responses in different developmental context and species (L350).

REVIEWERS' COMMENTS

Reviewer #1 (Remarks to the Author):

The authors have addressed all my comments and questions. The revised manuscript is significantly improved and ready for publication.

Reviewer #2 (Remarks to the Author):

All my points have been revised accordingly. I have no further criticism except for the figure design.

There is room for improvement. For example, the labels for the genotypes in Figure 1 B, E, H, Figure 3E, and Figure 4D and the GO terms in Figure 2D and E will be too small to read when made to the actual publication size.

In addition, the overall design of the box plots needs to be improved, such as the thickness of the lines on the X and Y axes and the letter for the Kruskal-Wallis test.

Reviewer #3 (Remarks to the Author):

The points I brought up have been all addressed. I have just a few minor points for the new parts.

Text

Line 200, The list of 444 genes identified in both ChIP studies as B-type ARRAs could be an interesting information for cytokinin researchers. I would propose to add this information as Supplemental file.

Line 22. Point at the end of the sentence

Line 222, Col0 is Col-0

Line 336, Riboside takes a lower case "r"

Line 375, "only in the dark for tomato" is unclear; Arabidopsis was not in the dark? Or tomato only in the dark but not in the cold? Please clarify

Line 391, 12-d-old

Line 392, Mock takes a lower case “m”; empty space before M (molar)

Line 454; “GO CK categories to filter down gene lists ...” the context and purpose is unclear. Filter (down?) which gene lists for what purpose?

Figures

The model shown in Fig. 5 has been improved, the signaling pathways and their activities under the different conditions are easier to follow. In the legend some more information should be given about the use of different colours, different type setting, size of letters etc. What does it all mean? In the drawing, the names of some factors (ARRs) do not fit in the respective circle, that can be improved. AHK2/AHK3 are signaling proteins like others but are shown above/below an arrow and not in a circle. That is why?

Response to reviewer comments for Gautrat et al.

We have taken all comments and written our response in green below each point.

REVIEWERS' COMMENTS

Reviewer #1 (Remarks to the Author):

The authors have addressed all my comments and questions. The revised manuscript is significantly improved and ready for publication.

Reviewer #2 (Remarks to the Author):

All my points have been revised accordingly. I have no further criticism except for the figure design.

There is room for improvement. For example, the labels for the genotypes in Figure 1 B, E, H, Figure 3E, and Figure 4D and the GO terms in Figure 2D and E will be too small to read when made to the actual publication size.

In addition, the overall design of the box plots needs to be improved, such as the thickness of the lines on the X and Y axes and the letter for the Kruskal-Wallis test.

We have not yet modified the figures at this point because it would probably be much more effective to receive any requests and instructions from the publisher or copy-editor, consistent with the journal policies and do this labour-intensive exercise (if necessary at all) in one action.

Reviewer #3 (Remarks to the Author):

The points I brought up have been all addressed. I have just a few minor points for the new parts.

Text

Line 200, The list of 444 genes identified in both CHIP studies as B-type ARRs could be an interesting information for cytokinin researchers. I would propose to add this information as Supplemental file.

Done. We have included this as the new Supplementary Table 2. We renamed the other tables accordingly and updated the text.

Line 22. Point at the end of the sentence

The reviewer probably meant line 221: we added a point.

Line 222, Col0 is Col-0

Done.

Line 336, Riboside takes a lower case "r"

Done.

Line 375, “only in the dark for tomato” is unclear; Arabidopsis was not in the dark? Or tomato only in the dark but not in the cold? Please clarify

Done. This was indeed unclear, and tomato seedlings were in the dark but not in the cold

Line 391, 12-d-old

Done.

Line 392, Mock takes a lower case “m”; empty space before M (molar)

Done.

Line 454; “GO CK categories to filter down gene lists ...” the context and purpose is unclear. Filter (down?) which gene lists for what purpose?

Done. This is now explicit what was done and for what purpose.

Figures

The model shown in Fig. 5 has been improved, the signaling pathways and their activities under the different conditions are easier to follow. In the legend some more information should be given about the use of different colours, different type setting, size of letters etc. What does it all mean?

Done. Colours are now described in the legend.

In the drawing, the names of some factors (ARRs) do not fit in the respective circle, that can be improved. AHK2/AHK3 are signaling proteins like others but are shown above/below an arrow and not in a circle. That is why?

We put AHK2/3 in boxes (this was an inconsistency in the design that is now solved) and fit everything in the circles now to the best of our abilities.